# Superconducting gap of H₃S measured by tunnelling spectroscopy

Feng Du[1✉], Alexander P. Drozdov[1], Vasily S. Minkov[1], Fedor F. Balakirev[2], Panpan Kong[3,4], G. Alexander Smith[2], Jiafeng Yan[5], Bin Shen[6], Philipp Gegenwart[6] & Mikhail I. Eremets[1]

Several hydrogen-rich superconductors have been found to show unprecedentedly high critical temperatures[1–4], stimulating investigations into the nature of the superconductivity in these materials. Although their macroscopic superconducting properties are established[1,5–7], microscopic insights into the pairing mechanism remains unclear. Here we characterize the superconducting gap structure in the high-temperature superconductor H₃S and its deuterium counterpart D₃S by performing tunnelling spectroscopy measurements. The tunnelling spectra reveal that H₃S and D₃S both have a fully gapped structure, which could be well described by a single s-wave Dynes model, with gap values 2Δ of approximately 60 meV and 44 meV, respectively. Furthermore, we observed gap features of another likely H-depleted H$_x$S superconducting phase in a poorly synthesized hydrogen sulfide sample. Our work offers direct experimental evidence for superconductivity in the hydrogen-rich superconductor H₃S from a microscopic perspective. It validates the phonon-mediated mechanism of superconducting pairing and provides a foundation for further understanding the origins of high-temperature superconductivity in hydrogen-rich compounds.

The discovery of the hydrogen-rich superconductor H₃S with a superconducting transition temperature $T_c$ of about 200 K has spurred the search for room-temperature superconductivity in hydrogen-rich compounds[1]. Despite the substantial challenges of high-pressure experiments on tiny samples, the macroscopic superconducting properties of hydrides have been well characterized, including electrical resistance[1,8–10], magnetization[6,7,11] and upper critical field[5]. However, there is limited experimental insight into the superconducting pairing mechanism in these materials. Although strong electron–phonon interactions and high-frequency phonons have been proposed theoretically as critical factors for the formation of Cooper pairs in hydrogen-rich superconductors[12], experimental validation has remained elusive. Furthermore, theoretical calculations of the superconducting parameters vary depending on the approach used[13–19]. Thus, a quantitative determination of the superconducting gap size and symmetry is essential, as it will provide an experimental basis to complete theoretical understanding and enable further predictions of superconductivity in hydrides.

Experimentally determining the superconducting gap under high pressure, particularly in a diamond anvil cell environment, is extremely challenging. Conventional techniques, such as angle-resolved photoemission spectroscopy and scanning tunnelling microscopy, cannot be used under high-pressure conditions. Although Capitani et al. used infrared reflectance spectroscopy to detect the superconducting gap and phonon frequencies of H₃S (ref. 20), further reluctance and absorption contributions from the diamond anvils and the NaCl insulating layer complicate the interpretation of the results. Andreev reflections have been observed in cerium hydrides[21], but the weak signal (approximately 2%) is insufficient to reliably extract information about the superconducting gap. Recently, Du et al.[22] developed a planar tunnel junction method, allowing for precise measurement of the superconducting gap structure under megabar pressure. Despite this advancement, the in situ chemical synthesis of planar tunnel junctions for hydrogen-rich superconductors with high purity and good homogeneity under high-pressure environment remains challenging.

In this work, we have successfully synthesized planar tunnel junctions of H₃S and D₃S using S + H₂ and S + D₂ precursors, respectively. Through tunnelling spectroscopy, we demonstrate that both compounds exhibit fully gapped structure, with gap values 2Δ of approximately 60 meV and 44 meV (obtained by fitting with the Dynes model), respectively. Furthermore, tunnelling spectra in the inhomogeneous sample synthesized using S and ammonia borane (NH₃BH₃) as an alternative hydrogen source reveal extra gap features from another likely H-depleted H$_x$S superconducting phase.

## Synthesis of planar tunnel junctions

The synthesis process of the planar tunnel junction for H₃S in a diamond anvil cell is schematically illustrated in Fig. 1a. A small piece of sulfur sample surrounded by hydrogen gas was compressed between two opposing diamond anvils after gas loading and subsequent

[1]Max Planck Institute for Chemistry, Mainz, Germany. [2]National High Magnetic Field Laboratory, Los Alamos National Laboratory, Los Alamos, NM, USA. [3]Beijing National Laboratory for Condensed Matter Physics, Institute of Physics Chinese Academy of Sciences, Beijing, China. [4]School of Physical Sciences, University of Chinese Academy of Sciences, Beijing, China. [5]Department of Physics, Institute of Nano Science and Technology, Hanyang University, Seoul, South Korea. [6]Experimental Physics VI, Center for Electronic Correlations and Magnetism, University of Augsburg, Augsburg, Germany. ✉e-mail: feng.du@mpic.de

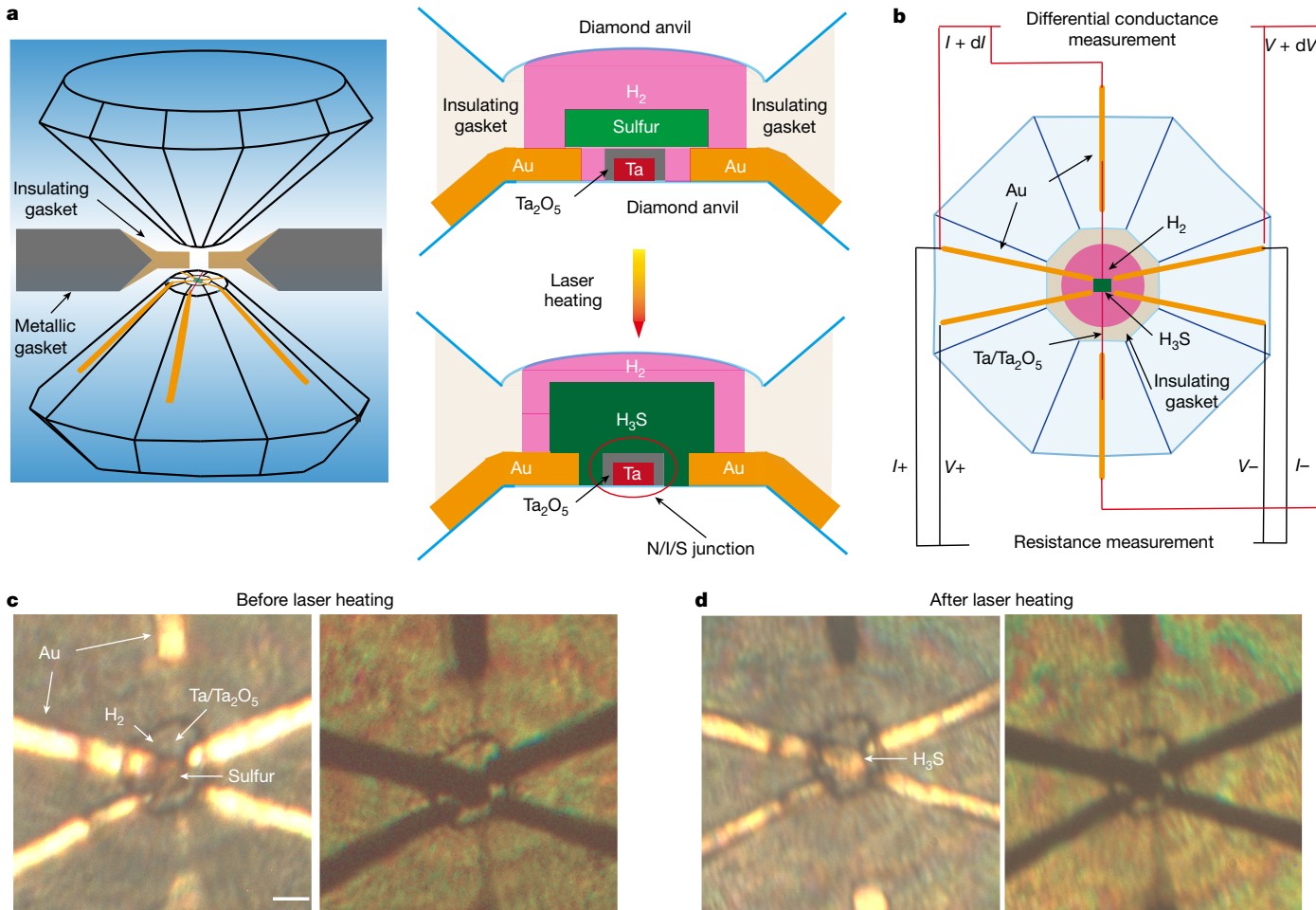

**Fig. 1 | Synthesis of planar tunnel junctions. a**, Schematic of planar tunnelling junction synthesis between two opposing anvils. Electrical leads are deposited on the bottom anvil and isolated from the metallic Re gasket with CBN/MgO/CaO insulating material. A rectangular sample (green) is loaded above the electrical leads. Detailed views of the region between the anvil tips before and after laser heating are shown from the side perspective. Gold leads are shown as thick orange lines and tantalum lead with Ta oxidized layer are coloured red and grey, respectively. The initial sulfur and synthesized $H_3S$ are marked in light green and dark green, respectively, surrounded by $H_2$ (pink). The Ta/Ta$_2$O$_5$/H$_3$S junction areas are indicated by the red oval. **b**, Schematic description of the differential conductance and resistance measurements from the top view of the region between the anvil tips. **c,d**, Optical microscope images (reflection and transmission) through the top diamond anvil of $H_3S$-S1 before (**c**) and after (**d**) laser heating at 158 GPa. Scale bar, 10 μm.

pressurizing. A roughly 15-nm tantalum thin film was deposited in the centre of the bottom anvil tip and an oxide layer was grown directly over the film to serve as normal metal (N) and insulating barrier (I) components of the N/I/S tunnel junction, respectively (detailed preparation of the electrical leads is provided in Extended Data Fig. 1). The key challenge to overcome in determining the superconducting gap in hydride materials is the synthesis of high-purity samples with only limited amounts of hydrogen available clamped between diamond anvils. Because planar tunnelling spectroscopy is not an atomic-resolution local probe such as a scanning tunnelling microscope and the measured tunnelling conductance is a sum of all components at the contact area, the presence of other phases in a multiphase sample can severely distort the gap feature of the desired phase. To ensure the necessary amount of hydrogen gas for complete chemical reaction, we made a cavity with a depth of about 1 μm and a diameter of about 30 μm on the top anvil with a focused ion beam machine (Extended Data Fig. 1d). Sulfur and hydrogen gas precursors were reacted using in situ laser heating, synthesizing $H_3S$ to serve as the superconducting component of the N/I/S junction. Then, the N/I/S junction was synthesized between two anvil tips, marked by the red oval in Fig. 1a. To prevent destruction of the

insulating barrier during laser heating, pulsed laser heating was implemented and long exposure times at the junction area were avoided.

The optical images of $H_3S$-S1 before and after laser heating are presented in Fig. 1c,d. Notably, we added two more gold leads to form a 'double' four-terminal configuration, which allows measurement of both the electrical resistance and the tunnelling spectra of the sample, as illustrated in Fig. 1b. The crystal structure of the synthesized samples was characterized by X-ray powder diffraction. Thus, we have characterized the synthesized samples from three aspects: electrical resistance, crystal structure and tunnelling spectroscopy.

In this work, we synthesized three high-purity $H_3S$ samples ($H_3S$-S1 (158 GPa), $H_3S$-S2 (151 GPa) and $H_3S$-S3 (161 GPa)) and one high-purity $D_3S$ sample (160 GPa)) using elemental sulfur and $H_2$ or $D_2$ gas. We also synthesized one hydrogen sulfide sample ($H_3S$-S4 (172 GPa)) using sulfur and ammonia borane as an alternative hydrogen source, which show multiphase features. Samples $H_3S$-S1 and $H_3S$-S4 were characterized by electrical resistance, X-ray diffraction and tunnelling spectroscopy, $H_3S$-S2 and $D_3S$ by X-ray diffraction and tunnelling spectroscopy and $H_3S$-S3 by tunnelling spectroscopy only.

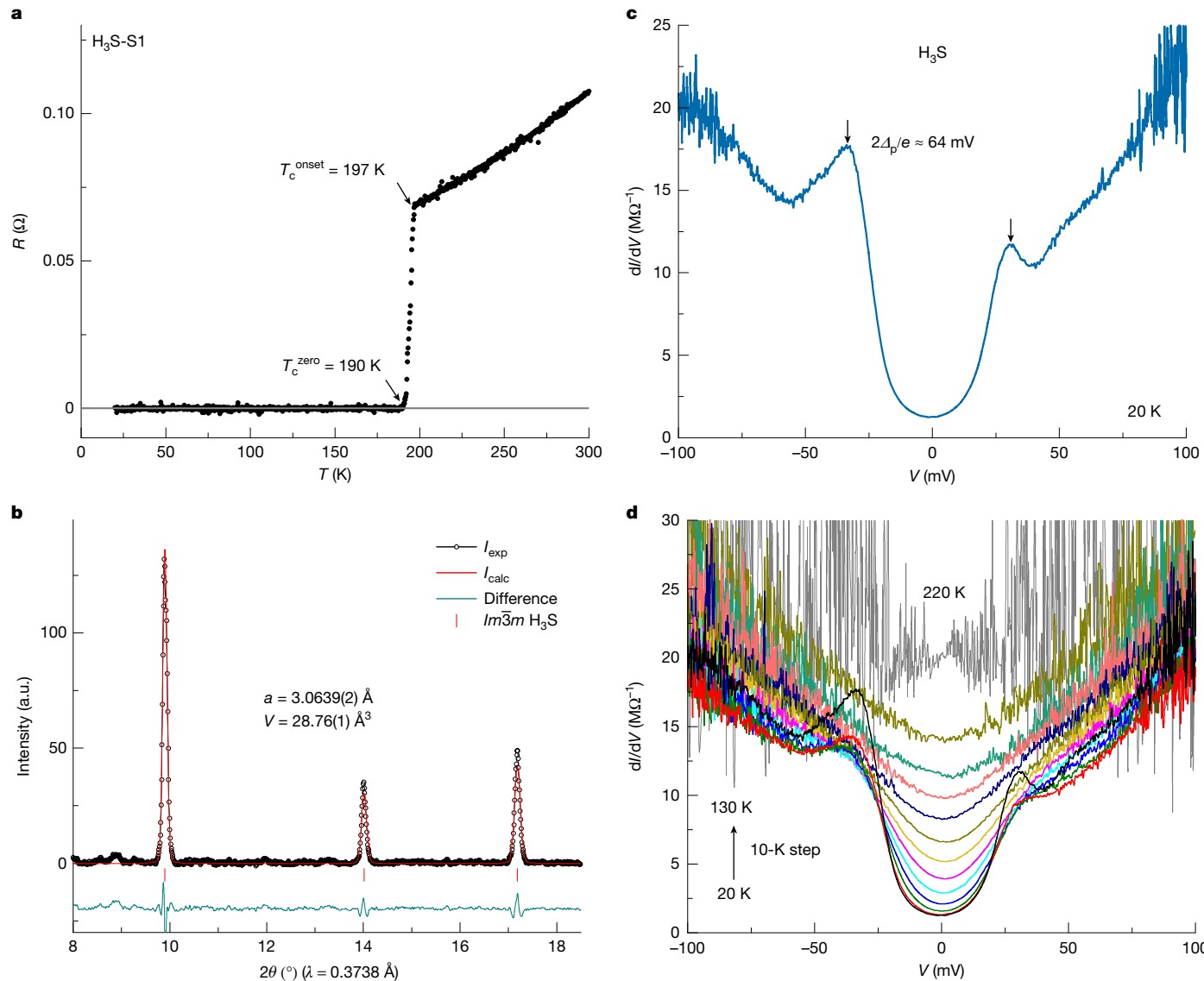

**Fig. 2 | Superconducting gap of H$_3$S. a**, Temperature dependence of the electrical resistance of H$_3$S-S1. **b**, X-ray powder diffraction pattern of H$_3$S-S1 (black data points) and the Rietveld refinement of the $Im\overline{3}m$ phase (red curve). **c**, Tunnelling spectra of H$_3$S-S1 measured at 20 K. Black arrows indicate positions of quasiparticle peaks. **d**, Temperature dependence of the tunnelling spectrum for H$_3$S-S1, measured from 20 K (black curve) to 130 K (olive curve) with 10-K steps and 220 K (grey curve, shifted by −7 M$\Omega^{-1}$ for comparison). a.u., arbitrary units.

## Superconducting gap of H$_3$S

Before exploring the superconducting gap features, we examine the resistance and crystal structure properties of H$_3$S-S1. As shown in Fig. 2a, the temperature dependence of electrical resistance exhibits a sharp drop at $T_c^{onset}$ = 197 K and then reaches zero at $T_c^{zero}$ = 190 K, indicating the formation of the superconducting phase in H$_3$S. The X-ray powder diffraction patterns (Fig. 2b and Extended Data Fig. 2) show the $Im\overline{3}m$ crystal structure with no further diffraction peaks through the sample, suggesting a high-purity single-phase sample of H$_3$S.

To characterize the superconducting gap features of H$_3$S, we conducted differential conductance measurements across the junctions at 20 K. Figure 2c shows the tunnelling spectra for H$_3$S-S1 at 158 GPa. A pair of coherence peaks emerges in the tunnelling spectra at symmetric positions ($\Delta^p_{H3S-S1} \approx \pm 32$ meV) relative to the Fermi energy, whereas the differential conductance within the energy below $\pm\Delta^p_{H3S-S1}$ drops to nearly zero, exhibiting a U-shaped structure, characteristic of the nodeless superconducting gap structure. Notably, the intensity of quasiparticle peaks at positive and negative

energies in tunnelling spectra shows a particle–hole asymmetry, which may be because of the energy-dependent transmission of tunnel electrons[23].

The temperature variation of tunnelling spectra for H$_3$S-S1 is shown in Fig. 2d. On warming, the height of the coherence peaks is continuously suppressed before fading into the background, and the bottom of the spectrum is elevated, owing to large thermal smearing at high temperatures. To quantitatively clarify the temperature evolution of the superconducting gap, we fitted the tunnelling spectra, which are normalized through division by the parabolic background fitted with Brinkman–Dynes–Rowell (BDR) model[24] (Extended Data Fig. 3), with a single $s$-wave Dynes model[25] (Methods). The parabolic background arises from the metal(N)/barrier(I)/metal(N) tunnelling process in the normal state[24,26] (Methods). As shown in Fig. 3a, tunnelling spectra at different temperatures are in good agreement with fits to the Dynes model, except the asymmetry of the intensity of quasiparticle peaks at low temperatures. The temperature evolution of the simulated gap values is summarized in Fig. 3b, which is in good agreement with the Bardeen–Cooper–Schrieffer (BCS) theory[27] (pink curve), and the gap

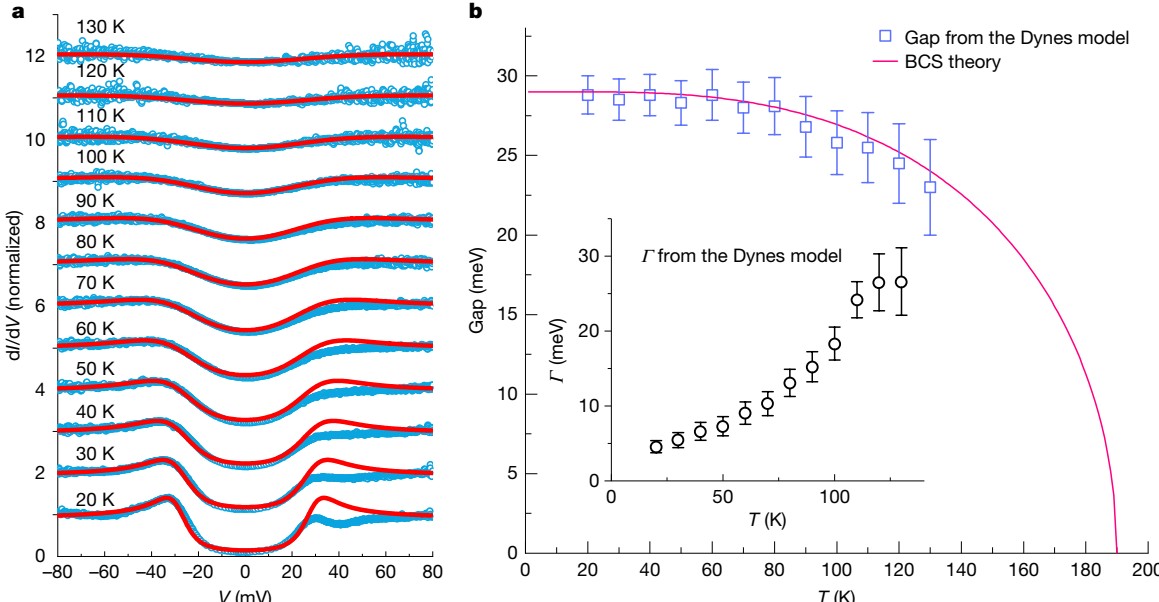

**Fig. 3 | Temperature evolution of normalized tunnelling spectra and comparison with the Dynes model. a**, Normalized tunnelling spectra (blue data points) and comparison with the Dynes model (red curves) at different temperatures. Spectra above 20 K are offset vertically for clarity. To minimize the influence of energy-dependent transmission in determining the gap value, the normalized tunnelling spectra were fitted using data at negative bias. **b**, Extracted gap and $\Gamma$ values from the Dynes model at different temperatures. Error bars are from fitting errors of the Dynes model. The pink curve represents the temperature dependence of the gap value from BCS theory, for which $\Delta_0 = 29$ meV and $T_c = 190$ K are used.

value $\Delta_{H3S-S1} \approx 29$ meV is extracted. Above 130 K, the gap features of $H_3S$-S1 become indistinguishable from the parabolic background owing to large thermal smearing and quasiparticle broadening effects, and no reliable gap value can be extracted (Extended Data Fig. 4).

Also, we have measured tunnelling spectra for $H_3S$-S2 (151 GPa) and $H_3S$-S3 (161 GPa) samples at 2 K, as shown in Extended Data Figs. 5 and 6. Both samples also exhibit a fully open gap structure with gap values $\Delta_{H3S-S2} \approx 29$ meV and $\Delta_{H3S-S3} \approx 31.5$ meV (obtained by fitting with the Dynes model), respectively. Although the magnitude of the normal state conductance differs among the three samples, the gap symmetry and value are well matched, giving $\Delta_{H3S} \approx 30(\pm1.5)$ meV. The observed superconducting gap is attributed to the random orientation of crystal directions, as evidenced by the well-defined powder rings in the X-ray diffraction patterns (Extended Data Fig. 2), indicating a uniformly oriented powder sample. There is no clear signature of the in-gap states or any further quasiparticle peaks in the three samples, suggesting an isotropic gap feature of $H_3S$. Furthermore, we have studied the magnetic-field-dependent evolution of the tunnelling spectra for $H_3S$-S3, as shown in Extended Data Fig. 6. The effective suppression of coherence peaks under magnetic field confirms the superconducting origin of the gap features. The gap value at 9 T shows only a slight decrease owing to the high upper critical field[5]. Further measurements are desirable to study the gap evolution under higher magnetic fields.

## Superconducting gap of D₃S

After the characterization of the superconducting gap of $H_3S$, we turn to its deuterium counterpart $D_3S$ to investigate the isotope effect. As shown in Fig. 4a, the synthesized $D_3S$ sample exhibits the same $Im\bar{3}m$ crystal structure as $H_3S$. The tunnelling spectrum of $D_3S$ at 20 K is shown in Fig. 4b. There are two kinks at the energy position $\Delta^P_{D3S} \approx \pm25$ meV, corresponding to coherence peaks, whereas the gap structure shows the same U-shape as for $H_3S$. To extract the gap value of $D_3S$, we normalized the tunnelling spectrum through division by the parabolic

background fitted with BDR model[24] (dashed pink curve) and fitted it with the Dynes model, which gives $\Delta_{D3S} \approx 22$ meV. The smaller gap value in $D_3S$, together with the same fully gapped feature, confirms the phonon-mediated pairing mechanism in the $H_3S$ superconductor. Notably, at high bias around ±60 mV, there is a clear step-like structural feature, as indicated by the green arrows in Fig. 4b, which may correspond to inelastic scattering from a bosonic phonon mode of $D_3S$ with energy difference $\Omega_{D3S} = E$ (60 meV) $- \Delta_{D3S}$ (22 meV) around 38 meV.

## Superconducting gap features of multiphase hydrogen sulfide

As mentioned above, phase inhomogeneity is a common phenomenon in hydrogen-rich compounds. For instance, several new phases of hydrogen sulfide have been observed in X-ray diffraction experiments, in which crystal structures of some concomitant phases differ greatly from the $Im\bar{3}m$ phase, identified as $H_4S_3$, $H_{2.85\pm0.35}S_2$ and $H_{6\pm0.4}S_5$ (refs. 28,29). We find that some of the planar tunnelling spectra features can be attributed to different phases present in multiphase samples.

A representative example is shown in Fig. 5. The sample $H_3S$-S4 (172 GPa) shows a two-gap feature in the spectra at 2 K. Except for the coherence peak at about 30 meV, which corresponds to the superconducting gap of $H_3S$ (Fig. 5a,b), there is a kink at around 8 meV (Fig. 5a). The kink is suppressed with increasing temperature and disappears above 40 K, suggesting that it corresponds to a second concomitant superconducting phase, which we denote $H_xS$. Indeed, electrical resistance measurements reveal a second superconducting transition with $T_c$ ($H_xS$) around 40 K (Fig. 5c). The X-ray powder diffraction data show that the sample contains several different phases. As shown in Fig. 5d, as well as the regions characterized predominantly by $Im\bar{3}m$ $H_3S$ (red region) and $\beta$-Po sulfur (yellow region) phases, there is another blue region with unique X-ray powder diffraction patterns that cannot be described by either of these two phases (Fig. 5e,f). Given that the superconducting gap features of the junction area (dashed rectangle in Fig. 5d) consist of contributions from both $Im\bar{3}m$ $H_3S$ and $H_xS$ phases,

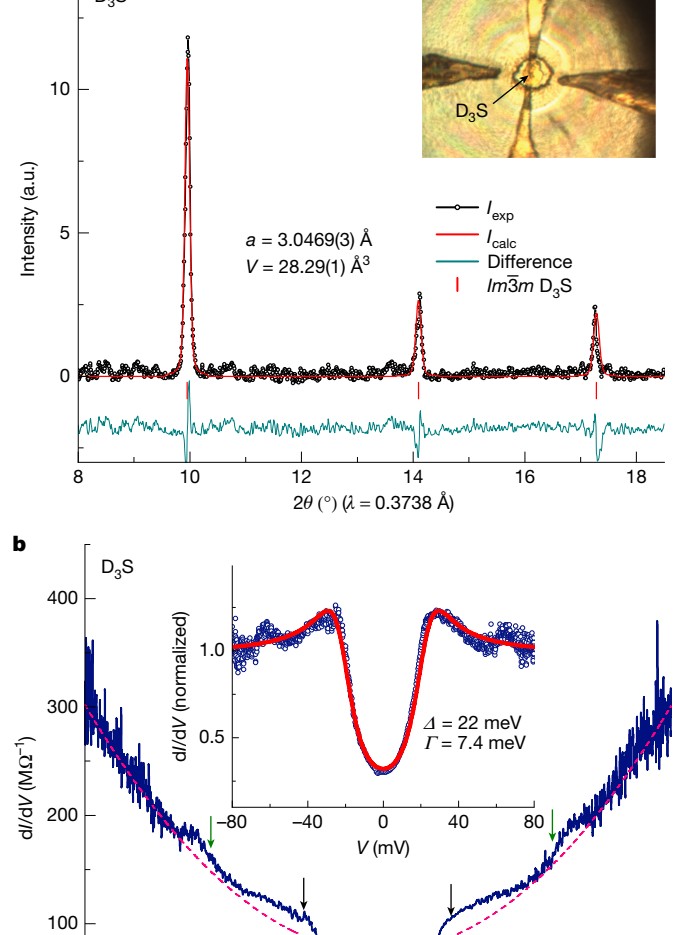

**Fig. 4 | Superconducting gap of D₃S. a**, X-ray powder diffraction pattern of $D_3S$ (black data points) and the Rietveld refinement of the $Im\bar{3}m$ phase (red curve). The inset is an optical image of the $D_3S$ sample. **b**, Tunnelling spectra of $D_3S$ (blue curve). Black and green arrows indicate the positions of quasiparticle peaks and step-like structures of $D_3S$, respectively. The dashed pink curve represents the parabolic background and the inset is the comparison between the normalized tunnelling spectra (data points) and the Dynes model (curve). a.u., arbitrary units.

it is likely that the X-ray powder pattern in this blue region is related to the $H_xS$ superconducting phase. The limited X-ray diffraction data do not allow us to resolve the crystal structure of the $H_xS$ phase. Further investigations into the crystal structure of $H_xS$ are desirable.

## Discussion and summary

As the order parameter, the superconducting gap is fundamentally related to the origin and nature of the superconducting coupling mechanism. Our observation of the superconducting gap by tunnelling spectroscopy provides unambiguous evidence for superconductivity in $H_3S$ and $D_3S$. The observed isotope effect and the inferred *s*-wave gap symmetry suggest that phonon-mediated Cooper-pair pairing is the dominant mechanism in $H_3S$. Unexpectedly, the observed superconducting gap value of $H_3S$ 30(±1.5) meV together with the $2\Delta/k_BT_c$(H₃S-S1) ratio of 3.54 are smaller than that

calculated by different theoretical approaches[18,30–32]. The discrepancies between experimental results and theoretical calculations require further detailed investigations on the origin of superconductivity in $H_3S$.

On the other hand, the observation of multigap features in inhomogeneous hydrogen sulfide sample reveals the existence of concomitant superconducting phases and call for a more comprehensive study of hydrides in which several superconducting phases are present. Finally, it would be insightful to apply this tunnelling technique to study superconducting gap structures in other hydride systems, such as metal superhydrides, as it would be helpful to explain the origins of high $T_c$ and to identify the pathways to new materials with higher $T_c$ and at lower pressure.

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

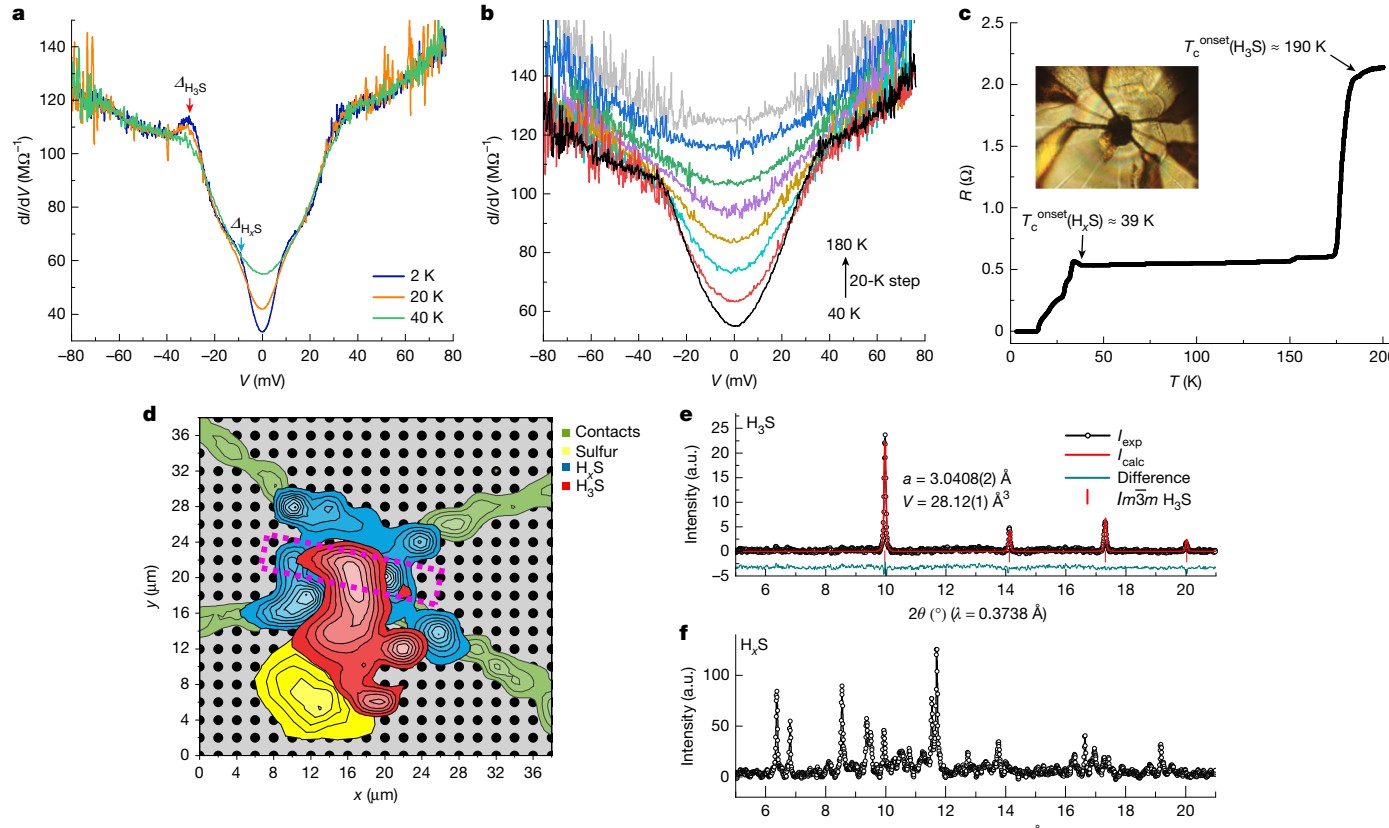

**Fig. 5 | Superconducting gap features of multiphase hydrogen sulfide sample. a**, Tunnelling spectra of H₃S-S4 measured at 2 K (blue curve), 20 K (orange curve) and 40 K (green curve). Arrows indicate positions of quasiparticle peaks of H₃S (red) and H$_x$S (cyan). **b**, Temperature dependence of tunnelling spectra for H₃S-S4, measured from 40 K (black curve) to 180 K (grey curve) with 20-K steps. The distortion of the fully gapped structure in the tunnelling spectrum of H₃S-S4 is the result of the relatively poor barrier quality and phase inhomogeneity, as analysed in Extended Data Fig. 7. **c**, Temperature dependence of electrical resistance for H₃S-S4. Arrows indicate $T_c$ of H₃S and H$_x$S. The inset is an optical image of the H₃S-S4 sample. **d**, Phase distribution in the multiphase sample H₃S-S4, reconstructed on the basis of X-ray powder

diffraction data. Contacts are indicated by the green area, with the sulfur, H$_x$S and H₃S regions shown in yellow, blue and red, respectively. The colour brightness indicates the relative phase content, with brighter colours corresponding to higher concentrations, as determined from the peak intensities at characteristic diffraction angles across different X-ray powder diffraction patterns. Grey points represent the points at which the experimental data were collected. The tunnel junction area is marked with a dashed rectangle. **e**, X-ray powder diffraction pattern of H₃S-S4 (black data points) and the Rietveld refinement of the $Im\bar{3}m$ H₃S phase (red curve). **f**, X-ray powder diffraction pattern of concomitant H$_x$S phase in H₃S-S4. a.u., arbitrary units.

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

## Methods

### Sample preparation inside the diamond anvil cell

Diamond anvil cells with culets of about 50 μm in diameter, bevelled to about 160 μm, were used in our experiments. 200-μm-thick rhenium gaskets were pre-indented to a thickness of 20 μm and then covered with an insulating film, prepared from a mixture of cBN and MgO/CaO/epoxy powder. A hole with a diameter of about the culet size was drilled using a laser before electronic leads were deposited on the diamond anvil in an ion beam sputtering machine. $Ta_2O_5$ barriers were created by oxidation of the Ta layer in a plasma machine under pure $O_2$ atmosphere under pressures of $10^{-1}$ mbar for 5 min at 350 W power. Polycrystalline sulfur samples were prepared from sulfur powder (purity of 99.999%, Alfa), which was first pre-compressed to thin plates with thickness about 1 μm and then cut into a rectangle shape with dimensions of about 10 μm × 5 μm. $H_2$, $D_2$ or $NH_3BH_3$ were used as sources of hydrogen or deuterium for the chemical reaction and pressure transmitting medium. Note that there may be imperfect contact between the sulfur and the electrodes in samples loaded with $H_2$ or $D_2$ before laser heating because a small amount of $H_2$ or $D_2$ may penetrate between them after the gas loading procedure. However, sufficient contact is made between the hydrogenated sample and the electrodes after laser heating owing to the expansion after hydrogenation of the sulfur sample. Using tunnelling spectroscopy for $H_3S$ and $D_3S$ samples at high pressures, we spent dozens of diamond anvil pairs exploring the best conditions for Ta oxidation, sample shape and thickness, gas loading and sample synthesis at high pressure.

### X-ray diffraction and pressure estimation

X-ray powder diffraction data were collected at beamline ID27 at the European Synchrotron Radiation Facility (ESRF) ($\lambda = 0.3738$ Å), with a beam spot size of $0.6 \times 0.6$ μm$^2$ (EIGER2 X CdTe 9M detector). $CeO_2$ powder was used as a reference sample for the calibration. Primary processing and integration of the data were carried out using DIOPTAS software and XDI software[33,34]. The indexing of X-ray powder diffraction patterns and refinement of the crystal structures were carried out using the GSAS[35] and EXPGUI[36] packages. The pressure was estimated using the Raman shift of stressed diamond anvils[37].

### Electrical resistance and differential conductance measurements

Electrical resistance measurements were performed in a four-probe configuration using a Keithley 6221 d.c. current source and a Keithley 2182A nanovoltmeter in delta mode. Differential conductance (d$I$/d$V$) ($V$) measurements were performed in a current-biased four-probe configuration using the Keithley 6221 current source and the Keithley 2182A nanovoltmeter in differential conductance mode. A d.c. current superimposed with a small delta current from a Keithley 6221 was applied across the junctions in the measurements, whereas the delta current was set to be as small as possible but also large enough to obtain an acceptable signal-to-noise ratio (10 nA for $H_3S$-S1, 5 nA for $H_3S$-S2, 4 nA for $H_3S$-S3, 100 nA for $H_3S$-S4, 100 nA for $D_3S$). The bias voltage and differential conductance were measured and calculated by a Keithley 6221. The two-wire electrical resistances of gold electrodes and tantalum strip are around 100 Ω and 30 KΩ, respectively. Further details on the operation of the differential conductance can be found in the Keithley manuals. All electrical measurements were performed in a Quantum Design Physical Property Measurement System.

### Data noise and temperature dependence of the background in tunnelling spectrum

There are three main sources of noise in the measurement of differential conductance. One is Johnson–Nyquist noise[38,39], which arises owing to thermal fluctuations and is proportional to temperature. The other two sources are flicker noise[40] and shot noise[41,42], which are the result of resistance fluctuations that generate a fluctuating voltage in the presence of a constant current and the discreteness of charge carriers, respectively. Because we use the same modulation delta current over the entire temperature and bias range for the same sample, flicker noise does not have a notable temperature and bias dependence. However, the Johnson noise increases with increasing temperature and the shot noise increases with increasing tunnel current in the circuit at high bias, which accounts for the reduced signal-to-noise ratio at elevated temperatures and high bias conditions.

The temperature dependence of the background arises from the temperature-dependent tunnelling conductance in the normal state, that is, metal/insulator/metal tunnelling. This phenomenon has been observed experimentally in normal metal tunnel junctions, such as Al/AlO$_x$/Al (refs. 43,44). Simmons[45] theoretically analysed the thermal influence on electron tunnelling by considering the smearing of the Fermi–Dirac distribution, which results in an increase of tunnelling conductance with temperature. Other theoretical models have been proposed to discuss the thermal influence on the tunnelling conductance, such as thermal-fluctuation-induced tunnelling[46]. Furthermore, such factors as the temperature dependence of the dielectric constant of the barrier, the thermal expansion of the barrier and the thermal activation across the barrier can also influence the tunnelling conductance at high temperatures.

### Dynes model

The Dynes model[25] describes the tunnelling density of states through an N/I/S tunnel junction, in which S is a BCS superconductor. The tunnelling density of states is given by the formula:

$$N_S(E) = N_0 \mathrm{Re} \left[ \frac{E + i\Gamma}{\sqrt{(E + i\Gamma)^2 - \Delta^2}} \right]$$

where $\Delta$ is the gap value, $\Gamma$ is the broadening parameter of the quasiparticle peaks, which includes contributions from both intrinsic quasiparticle recombination and extrinsic inelastic scattering, and $N_0$ is the density of states at the normal state. For fitting the data measured under magnetic fields, the Dynes model is modified by including the Zeeman splitting energy $\pm \mu_B B$, added to $E$. At zero temperature, the differential conductance (d$I$/d$V$)($V$) is directly proportional to the tunnelling density of states $N_S(E)$, whereas at finite temperatures, the thermal smearing effect should be considered, as follows

$$\frac{dI}{dV}(V) \propto \int_{-\infty}^{\infty} N_S(E) \left[ -\frac{\partial f(E + eV, T)}{\partial (eV)} \right] dE$$

in which $f$ is the Fermi–Dirac distribution.

### Data availability

Source data are provided with this paper. Any further data that support the findings of this study are available from the corresponding author on reasonable request.

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

# Article

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

**Acknowledgements** M.I.E., F.D., A.P.D. and V.S.M. are thankful to the Max Planck community for their support and U. Pöschl for the constant encouragement. We thank B. Wehinger, A. Pakhomova and M. Mezouar for assistance and support in using beamline ID27. We acknowledge the European Synchrotron Radiation Facility (ESRF) for provision of synchrotron radiation facilities under proposal numbers HC-5940 and HC-5483. The National High Magnetic Field Laboratory is supported by the National Science Foundation through NSF/DMR-2128556*, the State of Florida and the US Department of Energy.

**Author contributions** F.D. and M.I.E. designed the research; F.D. prepared the samples and performed the electrical measurements, with the help of A.P.D., V.S.M., P.K., F.F.B., G.A.S., B.S., P.G. and M.I.E.; F.D. and M.I.E. analysed the resistance and differential conductance data; F.D., V.S.M., J.Y. and G.A.S. measured X-ray diffraction and processed X-ray data. F.D., M.I.E., V.S.M., F.F.B. and G.A.S. wrote the manuscript, with input from all co-authors. We express our sincere condolences on the passing of M.I.E., who originally served as corresponding author and respectfully acknowledge his invaluable contributions.

**Funding** Open access funding provided by Max Planck Society.

**Competing interests** The authors declare no competing interests.

**Additional information**
**Correspondence and requests for materials** should be addressed to Feng Du.

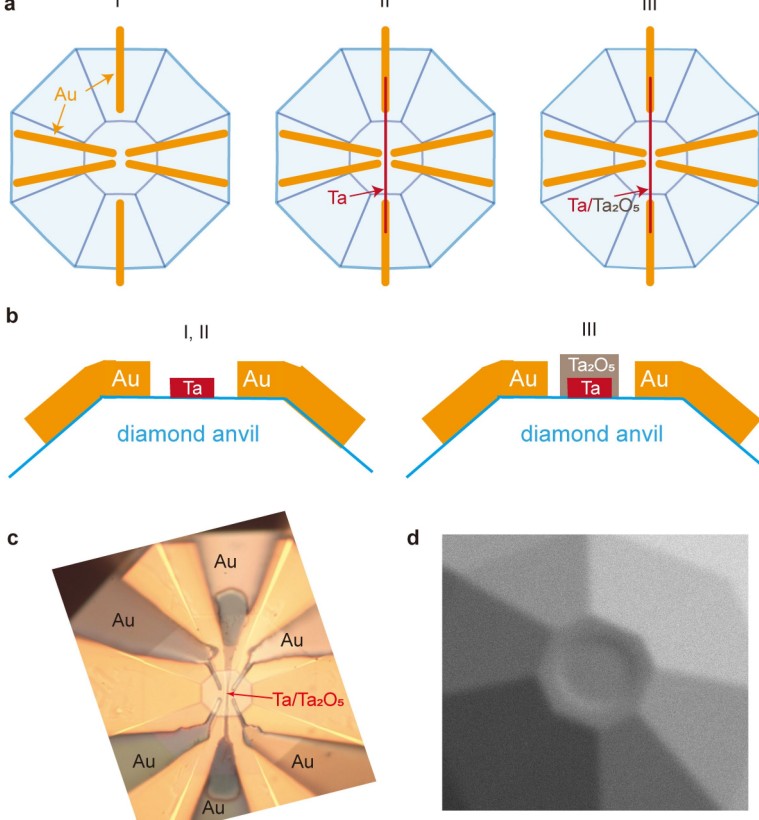

**Extended Data Fig. 1 | Preparation of electrical leads for tunnel junction.** Preparation of electrical leads onto diamond anvil from top view (**a**) and side view (**b**). Step 1: deposit six gold traces on the diamond anvil and connect them to the outer cooper wires (not shown), which will be connected to the measuring set-ups. Step 2: deposit Ta line and connect them to the outer up and down gold traces. Step 3: oxidize the surface of Ta line. **c**, Optical image of electrical leads deposited on the bottom anvil. **d**, Scanning electron microscope image of the top anvil after cavity fabrication by a focused ion beam machine.

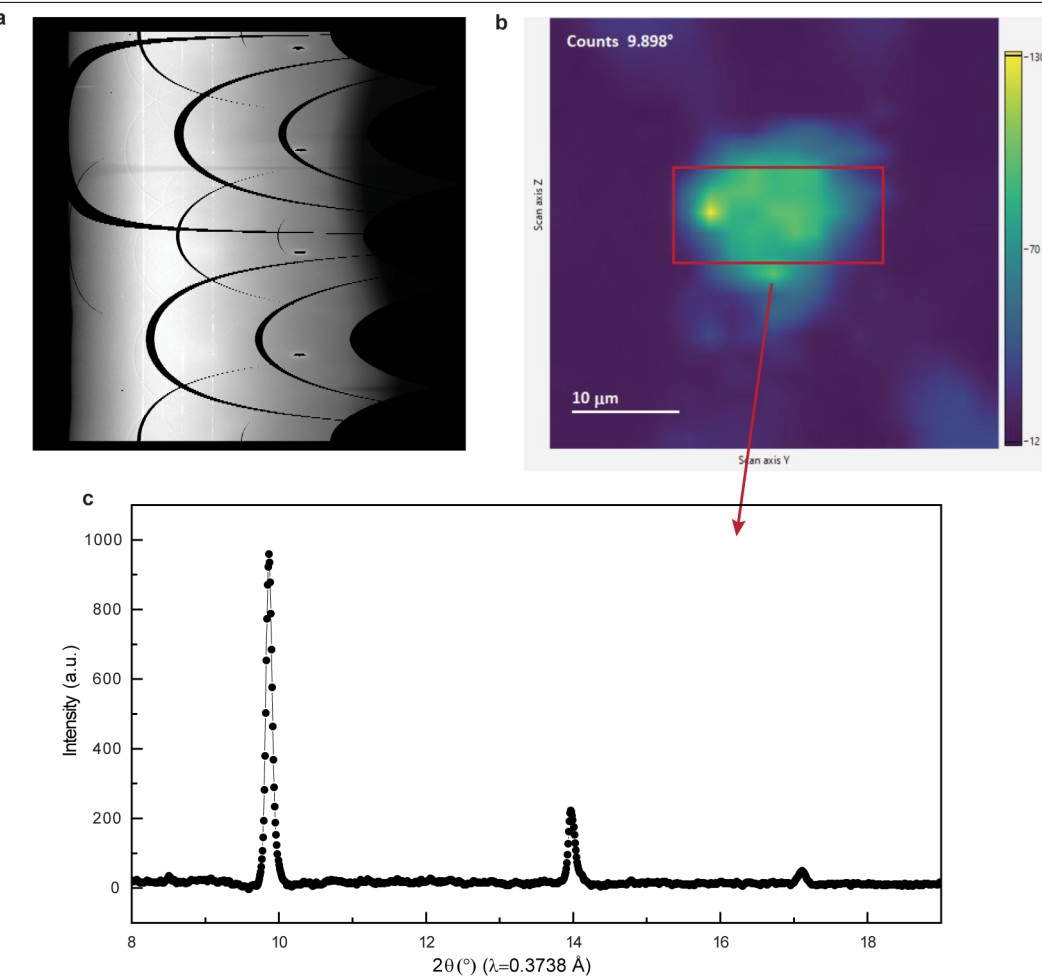

**Extended Data Fig. 2 | X-ray results of H₃S-S1. a**, Unrolled powder diffraction image of the data in Fig. 2b. **b**, X-ray diffraction map of H₃S-S1. The colour brightness varies with the intensity at the characterized diffraction angle around 9.9° at different data points. The tunnel junction area is marked with the red rectangle. **c**, Integrated X-ray powder diffraction pattern of the junction region of H₃S-S1.

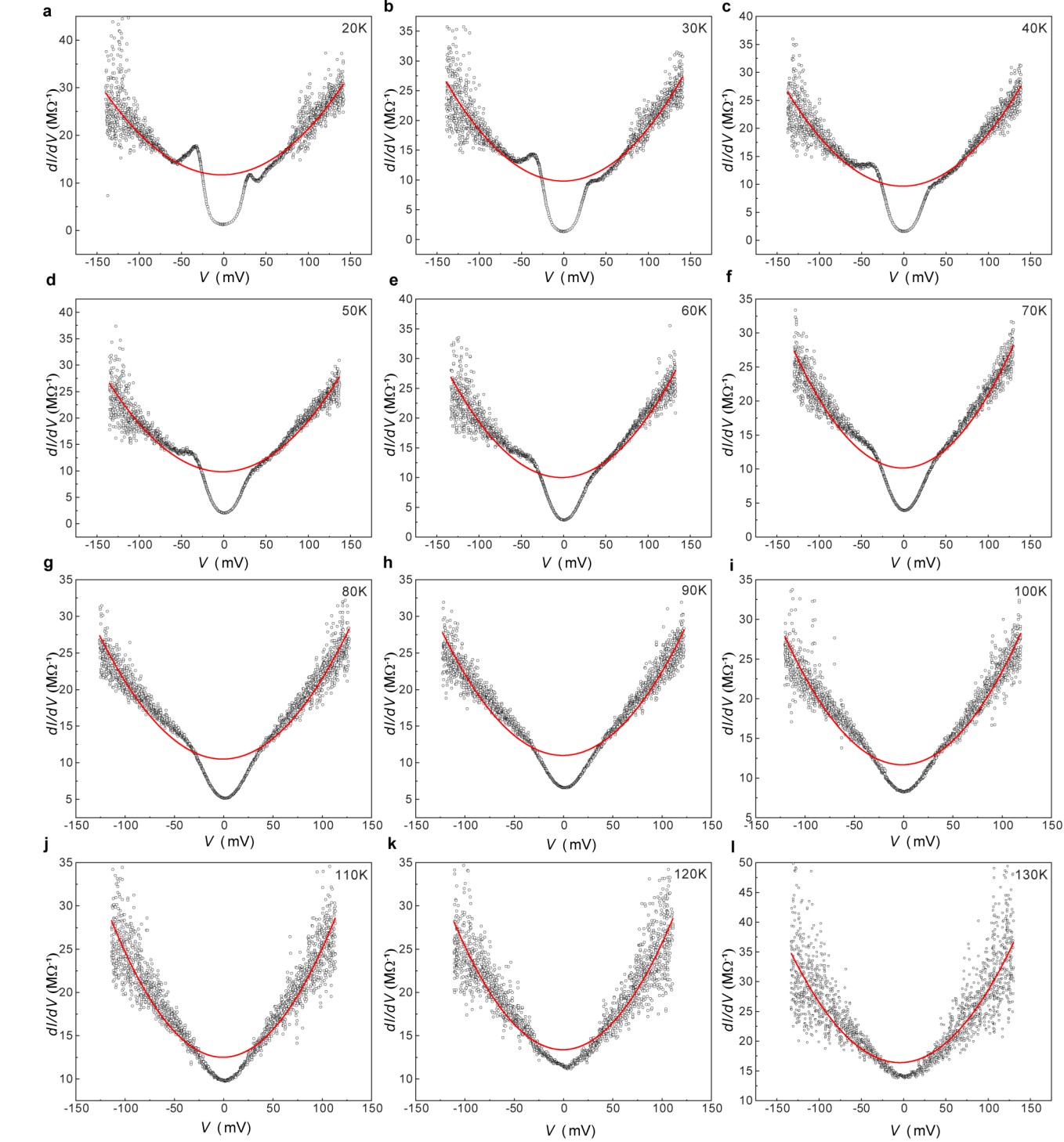

**Extended Data Fig. 3 | Fitting of background in tunnelling spectra of H₃S-S1. a–l**, Tunnelling spectra of $H_3S$-S1 (black data points) together with parabolic fits to the BDR model (solid red curve) at different temperatures.

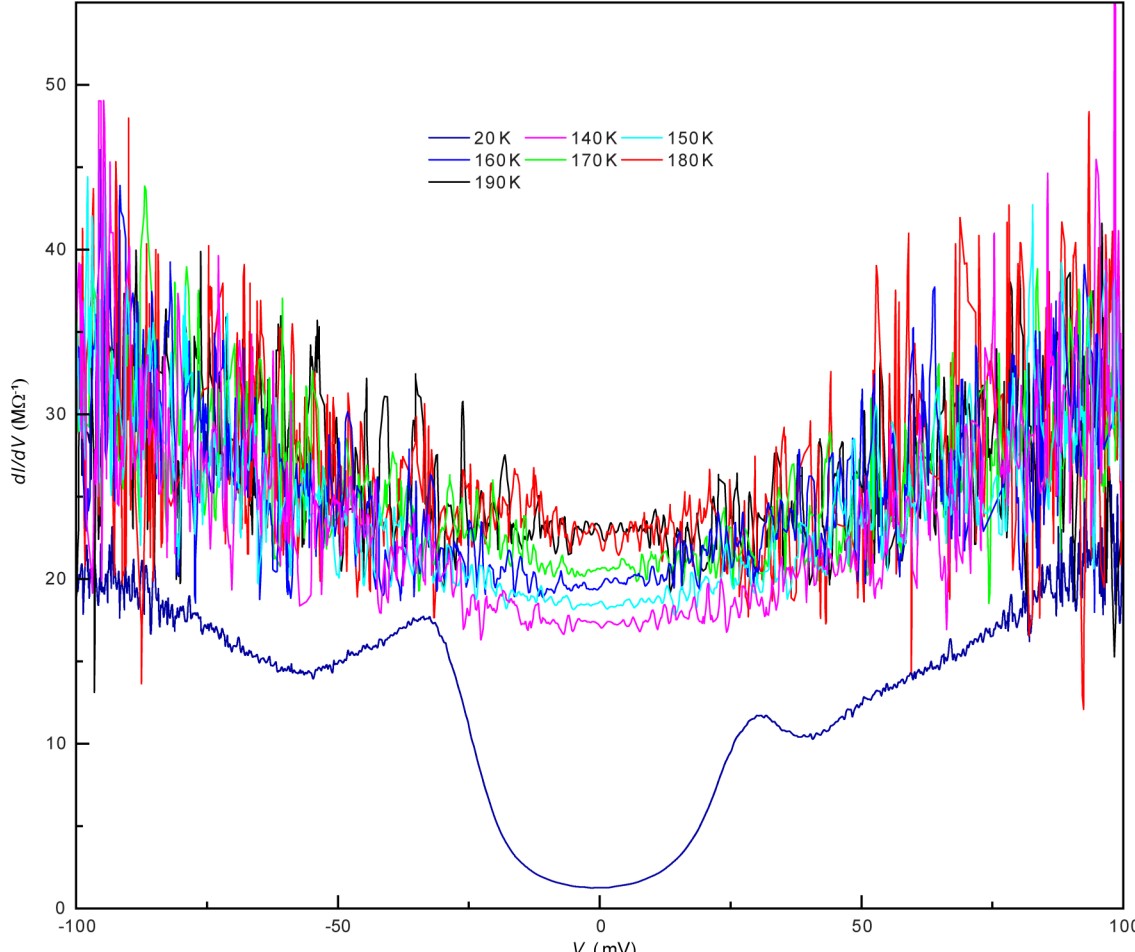

**Extended Data Fig. 4 | Tunnelling spectra of H₃S-S1 at high temperatures.** Tunnelling spectra of $H_3S$-S1 at 20 K and in the temperature range 140–190 K.

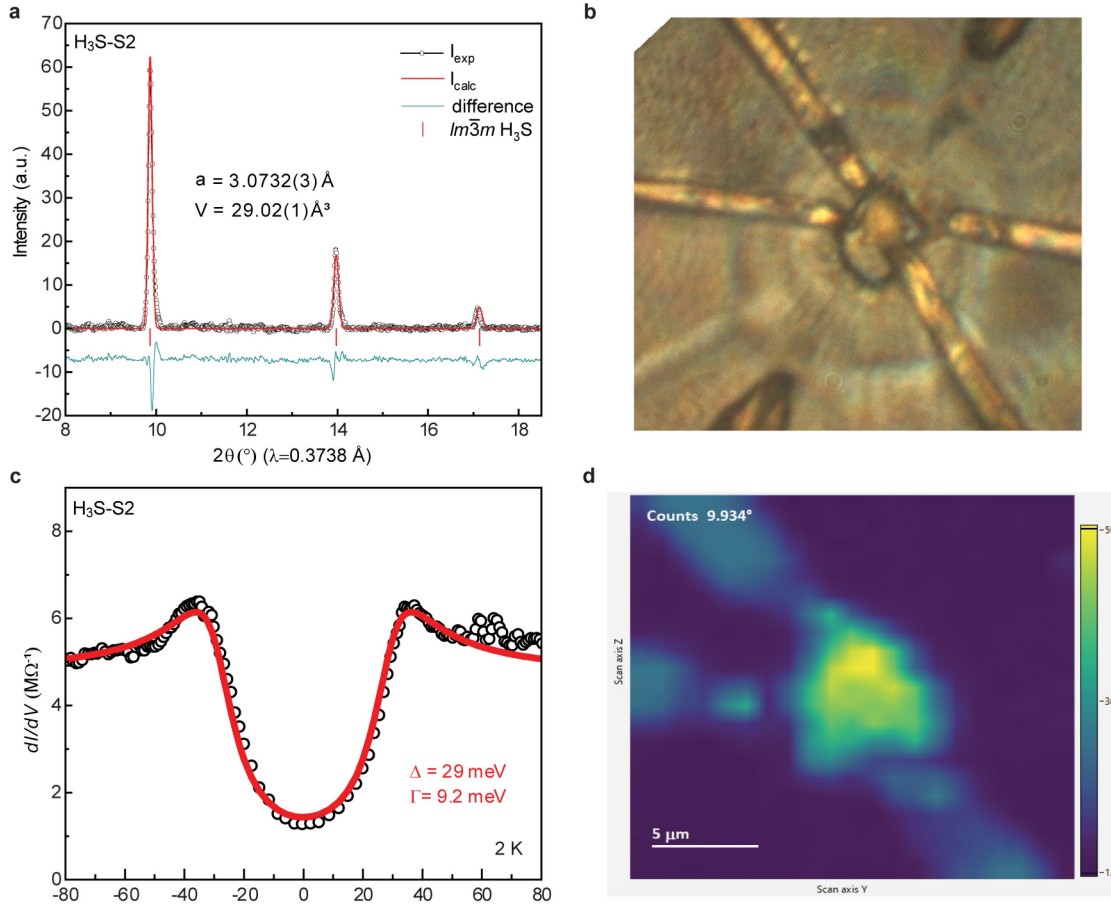

**Extended Data Fig. 5 | X-ray results and tunnelling spectra of H₃S-S2. a**, X-ray powder diffraction pattern of $Im\bar{3}m$ phase of H₃S-S2 (black data points) and the Rietveld refinement (red curve). **b**, Optical image of sample H₃S-S2. **c**, Tunnelling spectra of H₃S-S2 at 2 K (black data points) and simulation with the Dynes model (red curve). Δ and Γ values are extracted from the Dynes model. **d**, X-ray diffraction map of H₃S-S2. The colour brightness varies with the intensity at the characterized diffraction angle around 9.9° at different data points.

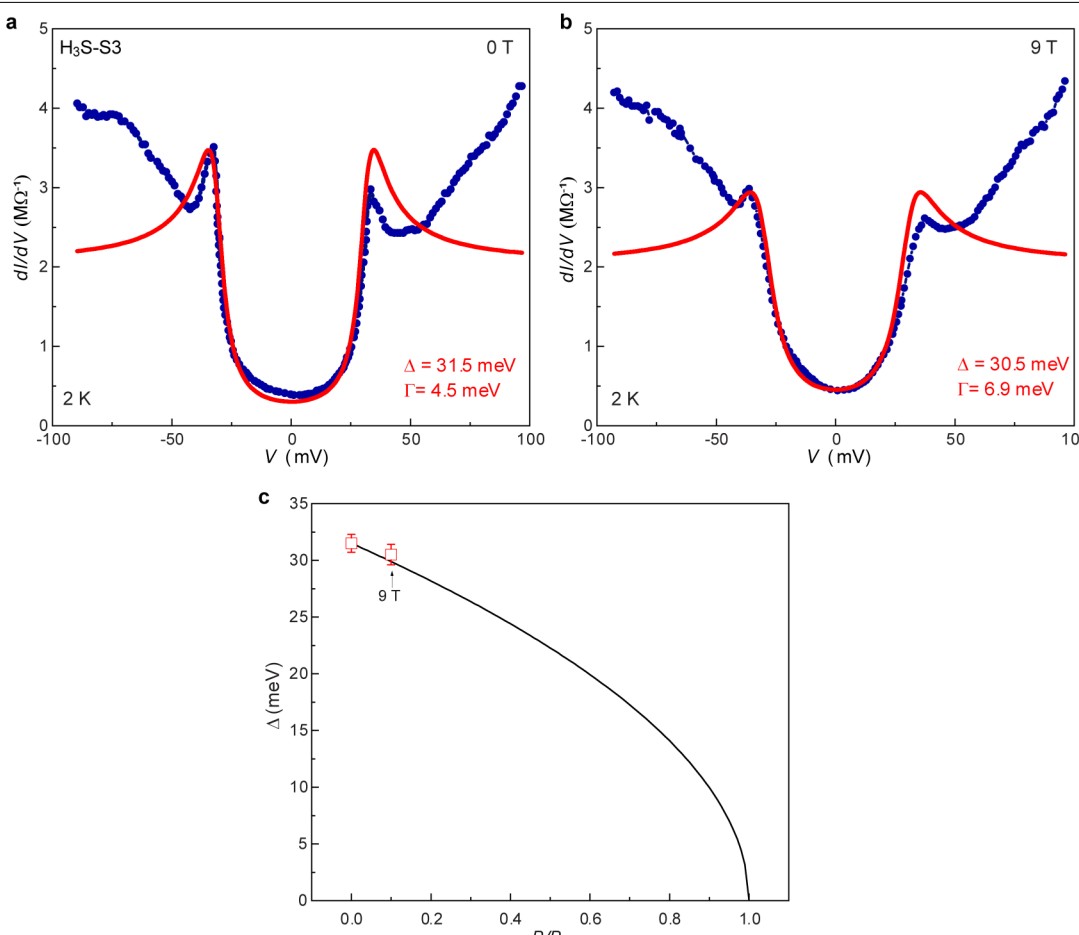

**Extended Data Fig. 6 | Tunnelling spectra results of H₃S-S3.** Tunnelling spectra of H$_3$S-S3 at 2 K (blue data points) fitted with the Dynes model (red curve), under external magnetic fields 0 T (**a**) and 9 T (**b**). Δ and Γ values are extracted from the Dynes model. The direction of the magnetic field is the out-of-plane direction. **c**, Gap evolution under magnetic fields (red points) and comparison with square root field dependence (black curve). Error bars are from fitting error of the Dynes model.

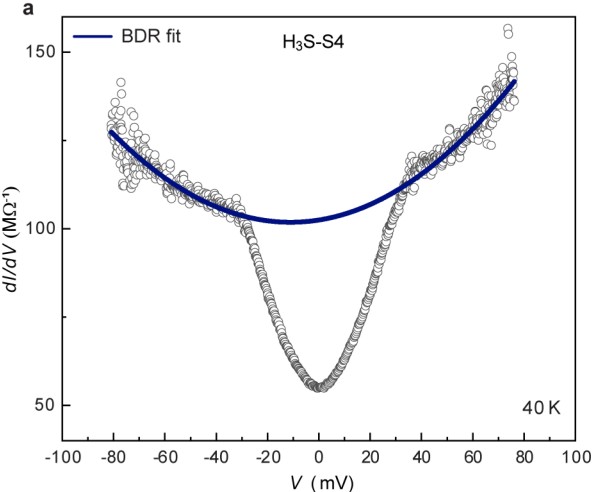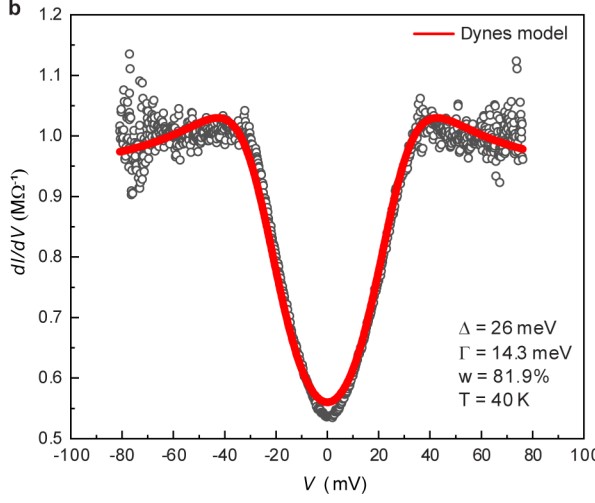

**Extended Data Fig. 7 | Analysis of superconducting gap in H$_3$S-S4.**
**a**, Tunnelling spectra of H$_3$S-S4 (black data points) together with parabolic fits
to the BDR model (curve) at 40 K. **b**, Normalized tunnelling spectra of H$_3$S-S4 at
40 K (black data points) and comparison with the two-component $s$-wave Dynes
model (curve): $N(E) = \omega N_S(E) + (1 - \omega)N_0(E)$, for which superconducting and
non-superconducting components are weighted by $\omega$ and $1 - \omega$, respectively.
The large gamma value indicates relatively poor barrier quality, whereas the
superconducting fraction $\omega = 81.9\%$ reflects phase inhomogeneity in H$_3$S-S4.
The slightly smaller gap value aligns with the lower $T_c$ of H$_3$S-S4 ($T_c^{offset} \approx 175$ K).