## [Peer Review file · Nature]

Superconducting gap of H₃S measured by tunneling spectroscopy

Corresponding Author: Dr Feng Du

This file contains all reviewer reports in order by version, followed by all author rebuttals in order by version. Parts of this Peer Review File have been redacted as indicated to remove third-party material.

Version 0:

Reviewer comments:

Referee #1

(Remarks to the Author)

Hydride superconductors have been the focus of the present field. The superconducting gap characteristics in high-pressure hydride superconductors has been a crucial issue. However, due to the challenges imposed by high-pressure experimental conditions, the conclusive experimental evidence is lacking now. In this work, the authors determine the superconducting gap characteristics of H₃S and D₃S employing high-pressure tunneling spectroscopy, providing insights into the superconducting mechanism of hydride superconductors. However, I think the present data quality is not enough to prove the conclusions, and the present version requires major revisions. Several important questions or aspects regarding the data and viewpoints presented in the manuscript should be confirmed and modified before the final publication:

1. How can the authors ensure sufficient contact between the sample and the electrodes under high pressure to achieve reliable electrical transport and tunneling spectra signals? If the S sample and electrodes are in close contact, it will indicate that there is minimal hydrogen content between them before heating. In this condition, how to ensure that the sample fully reacts to form H_xS after heating? Alternatively, if the sulfur sample is fully enveloped by hydrogen, how to achieve high-quality electrical transport and tunneling spectra signal?
2. In Fig. 2c, the manuscript attributes the asymmetry of the differential conductance curves to an asymmetric metal-oxide tunneling barrier. Does this imply that the Ta₂O₅ film has poor uniformity?
3. They used the Dynes model to fit the tunneling spectrum data in Fig. 3. What advantages does this model offer over the BTK model? I noticed that the barrier parameters are not fitted in this model. Does this make the fitting results less reliable compared to the BTK model?
4. Based on the electrical curves in Fig. 2, zero resistance can be maintained up to 190 K. But why does the tunneling spectrum signal exhibit such significant distortion due to thermal effects above 130 K?
5. The XRD patterns indicate good powder diffraction rings, suggesting that the sample have low crystallinity and thus exhibits isotropic properties. Could this result in the broadening of the differential conductance curve, a U-shaped curve with s-wave gap characteristics?
6. In Fig. 4b, how is the red dashed background line obtained? Do the authors use the differential conductance curve at the normal-state temperature as the background, or fit a parabolic curve as the background?
7. In Fig. 5a, a V-shaped differential conductance curve near zero bias voltage is observed, while a U-shaped differential conductance curve is also noted earlier in the manuscript. Is it reliable for the authors to determine the superconducting pairing symmetry based on the shape of the differential conductance curve?
8. In Fig. 5, the authors mention that the sample is mixed phases. How to rule out the possibility of sample layering, which could lead to the observation of a Josephson signal between the two sample layers?
9. The authors should supplement the evolution of the superconducting gap with magnetic field at different temperatures. This will give the information of the superconducting gap dependence on the magnetic fields.

10. The authors claimed they got superconducting gap value of 30 meV for H3S, but this value is much smaller than the predicted one (PRB 2015, 91, 060511(R)). How to explain this large discrepancy? In general, this method would give the higher superconducting gap due to the broaden of the tunneling caused by the scattering or thermal effects.

Referee #2

(Remarks to the Author)

Results:

Du et al. present electron tunnelling experiments on the high-temperature superconductors H3S and D3S. The study observes a gap in the electronic states consistent with the expectations for superconductors. The spectra show coherence peaks which are a fingerprint of superconductivity and a U-shaped profile indicating fully gapped superconductivity. The magnitude of the gap is an exact match with the expected value from weak-coupling BCS theory and the difference in gap magnitude for H3S and D3S is consistent with the isotope effect expected for electron-phonon superconductors.

Originality:

This is the first reliable report of electronic spectroscopy in the whole class of hydride high-temperature superconductors. Previous related work was limited to small signatures (~2%) of Andreev reflection on top of a complex background in CeH9 [1]. Other previous work by some of the co-authors on optical reflectivity was obscured by background contributions and is not generally accepted as a reliable [2]. The present work by Du et al. is successfully analysed as 100% quasiparticle tunnelling.

Significance:

The reliable observation of the superconducting gap in this report is highly significant for the field of hydride superconductivity and beyond. Since the discovery of high-temperature superconductivity in H3S, the whole field of hydride superconductivity has been under close scrutiny. In fact, the dialogue has left many observers in doubt whether superconductivity is the correct interpretation. In addition, academic integrity standards of the high-pressure community have been questioned.

Robustness of Conclusions:

The present study provides very solid evidence that superconductivity is the correct interpretation of the signatures observed in H3S and D3S. The observation of a gap in the electronic states combined with the drop in resistance observed here on the same samples rules out virtually all alternative interpretations. In addition, the observed coherence peaks are a fingerprint of superconductivity.

Data and Methodology:

The present manuscript presents highly reproducible data on multiple samples. If this can be complimented by an immediate and full publication of the raw data, it will also rebuild trust in academic integrity. I further encourage the authors to indicate the total number of pressure cells and samples prepared for tunnelling experiments and what variation of results has been observed in that wider set if applicable.

Given the wide significance and novelty of the results, I recommend publication of the manuscript in Nature once the authors have considered my comments.

I list a few more specific comments:

- 1) The authors use the Brinkman-Dynes-Rowell model to analyse the background and the Dynes model to analyse the superconducting gap. These models are appropriate.
- 2) The authors write that they “simulated” the Dynes model. Do the authors mean that they fitted the Dynes model to the data using a least-squares fit or similar? If not, they should explain how the uncertainties on Δ and Γ (Fig. 3b) have been obtained.
- 3) In the analysis of the Dynes model, the authors include quasiparticle broadening (Γ). The authors should clarify what origin they associate with the broadening as the original work by Dynes linked this to intrinsic quasiparticle recombination whilst in the present study it appears more likely to be linked to extrinsic factors.
- 4) The authors link the U-shape of the spectra to single-gap s-wave superconductivity. Indeed, the U-shape provides evidence for fully gapped superconductivity which, however, can be realised with other order parameters. Thus, the authors should clarify this argument. The authors might want to rephrase statements like “directly reveal the superconducting gap structure” which suggests k-space resolved spectroscopy.
- 5) Fig 2: To provide a reliable reference measurement, the authors must include a curve of the background in the normal state, i.e. well above 200K.
- 6) Fig2: The authors omit curves between 140K and 180 K – inside the superconducting state. I encourage the authors to include this data in the supplementary information for completeness even if the data cannot be analysed with the Dynes model due to large broadening.

7) The background shows a large temperature dependence for sample S1 but not S4. The authors should discuss the origin of the temperature dependence.

8) The author suggest thermal drift as a source of reduced energy resolution. However, the method indicates use of a PPMS which typically achieves temperature stability of 1mK over the full temperature range of this study. The authors should clarify this.

9) The data show a large increase of noise with increasing bias and increasing temperature for sample S1 and S4 but not for samples S2 and S3. The authors should comment on the origin of this noise.

I encourage the authors to include key characteristics of the setup in the methods or supplementary information. This should include the resistance of the tantalum strip and electrodes as this will allow readers to understand the methodology and expected noise level from Johnson noise and shot noise. Can the authors include details on the magnitude of the delta current used?

10) For sample S1 and S4, the authors observe asymmetry of the quasiparticle peaks whilst this is absent for S2 and S3. The authors suggest this to originate from an asymmetry of the barrier referring to [3,4]. However, I cannot find any model or explanation in these references that is able to make this link. In fact, the asymmetry of the junction is accounted for in the background fit using the BDR model as a horizontal shift of the minimum and appears to be small for S1. Thus, I ask the authors to clarify this argument.

11) The authors also present evidence for a new H-depleted H_xS compound and associate this with the use of ammonia borane as hydrogen donor. The strict association with ammonia borane appears to contrast earlier work of some of the co-authors and others who have reported phase-pure H₃S synthesised from sulphur and ammonia borane [5–7]. The authors may want to comment on this.

12) The authors claim that the suppression of the coherence peaks in a magnetic field of 9T confirms the superconducting origin. Whilst there is a transfer of density of states from the coherence peaks into the gap region, it appears that the centre of the coherence peaks shifts to higher energy. Can the authors comment on this aspect? Can the authors extract a magnitude of the gap and compare this with the expected suppression in magnetic field?

13) I suggest to specify the direction of magnetic field relative to the planar junction.

14) I recommend the authors to compare their work to that of Cao et al who observed Andreev reflection in CeH₉ [1]

15) Finally, I wonder whether the authors want to consider a more specific title potentially indicating the experimental nature of the work.

[1] Cao, Z.-Y. et al. Probing superconducting gap in CeH₉ under pressure. arXiv:2401.12682 [cond-mat.supr-con] (2024). (DOI:10.48550/ARXIV.2401.12682)

[2] Capitani, F. et al. Spectroscopic evidence of a new energy scale for superconductivity in H₃S. Nat Phys 13, 859–863 (2017). (DOI:10.1038/nphys4156)

[3] Du, F. et al. Tunneling Spectroscopy at Megabar Pressures: Determination of the Superconducting Gap in Sulfur. Phys. Rev. Lett. 133, 036002 (2024). (DOI:10.1103/PhysRevLett.133.036002)

[4] Wolf, E. L. Principles of electron tunneling spectroscopy. (Oxford University Press: Oxford, 2012)

[5] Minkov, V. S. et al. Magnetic flux trapping in hydrogen-rich high-temperature superconductors. Nature Physics 19, 1293–1300 (2023). (DOI:10.1038/s41567-023-02089-1)

[6] Minkov, V. S. et al. Magnetic field screening in hydrogen-rich high-temperature superconductors. Nat. Commun. 13, 3194 (2022). (DOI:10.1038/s41467-022-30782-x)

[7] Osmond, I. et al. Clean-limit superconductivity in Im₃m H₃S synthesized from sulfur and hydrogen donor ammonia borane. Phys. Rev. B 105, L220502 (2022). (DOI:10.1103/PhysRevB.105.L220502)

Referee #3

(Remarks to the Author)

I have reviewed the paper "Superconducting gap of high temperature superconductor H₃S" by F. Du and coworkers. In this paper is reported new data on the synthesis and characterization of sulphur hydride at extreme pressure with a particular focus on the measurement of the superconducting gap. This work confirms the observation of high T_c superconductivity in the Im-3m structural phase of H₃S and provides and measures a superconducting gap of about 30meV (reduced to 22meV with the isotopic substitution H to D).

The impression is that this work presents the most precise and complete characterization of H₃S which I have seen so far. I think that this is relevant because there have been controversial claims on other high pressure hydrides and the reproducibility (/improvement) of earlier measurements in H₃S is a significant fact. The paper is clear and the methodology appears to be cutting-edge. However, as I have a theoretical background, I will not comment on the experimental aspects of

this work.

From a theoretical point of view there is no (reasonable) doubt about the phononic nature of superconductivity in H₃S. In fact superconductivity was predicted before the experiments. The agreement between theoretical and experimental T_c is also quite good in all the many published works (I should add: within the typical theoretical/experimental errorbar). In fact, the agreement is almost too good, because different methods and approximations all lead to very similar values of T_c. For example it is generally accepted that anharmonic effects are important in this class of materials, as these have a major effects on the lattice dynamics. However even in the harmonic approximation T_c estimations are reasonable. It nails down to the fact that T_c is not the most sensitive property of phononic superconductors.

The superconducting gap is different. Its estimation can be affected by the use of different theoretical approximation (e.g. harmonic/anharmonic; BCS/Eliashberg/SCDFT ; isotropic/anisotropic). An accurate estimation of the superconducting gap is, for the theory community, a very valuable element to test the methods and possibly to improve them.

Clearly there are thousand of superconducting systems where these theories can be tested, but none of them has the properties of H₃S. H₃S is the most extreme available data point in terms of T_c and characteristic phonon frequency.

In this work it is reported that the experimental values of the gap are quite lower than in the theoretical simulations. I would place the most precise theoretical value for H₃S to about 36K using Eliashberg theory and anharmonic phonons. The disagreement with the experimental estimation of 30meV is certainly not large enough to doubt of the phononic mechanism, but it indicates a problem that, as the authors mention in the conclusion, should be addressed.

The explanation could be as trivial as a failure of the Dyson model used for the fitting and the comparison with an isotropic (clean limit) estimation of the gap. But this data could also be pointing to the necessity of including non trivial effects to our methods.

One thing that comes to my mind is the feedback effect of the superconducting condensation. In a system with an extremely high gap, its presence could modify the phonon spectrum at low temperature. This mechanism would affect the theoretical estimation of the gap, without affecting the estimation of T_c (being the transition of the II order the gap is zero at T_c).

In conclusion I have a good opinion of this work. An accurate confirmation of the SH3 measurements is certainly welcome. Most importantly I think that this work provides extremely valuable data which could be quite useful for the improvement of superconductivity theory.

Version 1:

Reviewer comments:

Referee #1

(Remarks to the Author)

The authors have made some improvements to the paper. However, they have not really answered my queries. I still have several issues regarding this work as follows:

1. In Supplementary Material Fig. S1, they provided an image of the diamond surface used for high-pressure tunneling spectroscopy measurements. I estimated the width of the Ta line on the diamond anvil surface to be approximately 2–3 microns. For tunneling spectroscopy measurements under ambient pressure, the tip is required to be at the nanometer scale. Do you think a micron-scale contact area can meet the experimental requirements? Will the contact area have a significant impact on the experimental results?
2. I'm still puzzled as to why the tunneling spectroscopy signal decays so rapidly in the temperature range between 130 K and 190 K, even under conditions of zero resistance. In the high-pressure tunneling spectroscopy experiments of elemental sulfur (PRL 133, 036002 (2024)), I didn't observe the similar phenomena. Although the authors mentioned that the signal quality significantly deteriorates above 130 K, according to the curve in Figure 3a, the tunneling signal is already no longer obvious above 110 K. Based on the article in Nature, 525, 73–76 (2015), H₂S exhibits a superconducting transition temperature near 100 K under similar pressure ranges. Therefore, is it possible that the Ta-Ta₂O₅ tunneling junction, spanning the entire sample, is detecting a mixture of signals from H₃S and H₂S? The zero-resistance observed in electrical transport measurements only proves the formation of a superconducting path of H₃S between the two gold electrodes. The powder XRD diffraction pattern provided by the authors only confirms the presence of pure-phase H₃S in one specific region, but it does not cover all the samples within the sample chamber. Thus, the accuracy of the bandgap information detected using this method remains open to discussion.
3. The resistance of an insulator changes exponentially or linearly with temperature, which should result in a faster rate of change at low temperatures. However, why does the contact resistance in the variable-temperature tunneling spectroscopy data presented in the paper show a slower rate of change at low temperatures and a faster rate of change at high temperature range?
4. Why do the tunneling spectroscopy curves in Figure 5 exhibit a V-shaped characteristic different from other DACs?

5. The tunneling spectroscopy curves presented by the authors in the high-pressure region generally have a poor signal-to-noise ratio, while the signal-to-noise ratio is better in the voltage range within the superconducting gap. Did the authors perform any fitting or signal processing on the curves to improve their quality?

Based on these questions, I think this work is not appropriate for publication in Nature in the present form. I would recommend further modifications are necessary.

Referee #2

(Remarks to the Author)

I recommend publication of the manuscript. The authors have addressed all my queries satisfactorily and I only have a single remaining comment (see below).

I suggest the authors to describe/check carefully, their data fitting routines. The fits shown in Fig. 3a seem much closer to the data for negative bias than for positive bias. I wonder whether only the data at negative bias has been fitted. If so, this should be clearly described and discussed. Whilst I don't expect that this has significant relevance for the main conclusions of the manuscript, the authors may want to be cautious to avoid any ambiguity.

Referee #3

(Remarks to the Author)

I believe the Authors have replied to all the points raised by the referees. I maintain my opinion that this paper is suitable for publication in Nature.

Version 2:

Reviewer comments:

Referee #1

(Remarks to the Author)

In this revised version, I think the authors have addressed most of my queries. I recommend publication of the manuscript.

Dear Editor,

Thank you for considering our manuscript for publication in Nature. We are grateful for the reviewers' insightful comments and valuable suggestions, which have helped us improve the quality of our work. In response, we have carefully addressed all critiques and incorporated the corresponding revisions into the manuscript. The changes are highlighted in red for your convenience.

Best regards,
The Authors

Response to the referees

We are very grateful to all the referees for reviewing our manuscript, as well as for the various comments and questions. We respond to all of the referees' comments in detail below, followed by a list of changes in the revised manuscript.

Referee #1 (Remarks to the Author)

“Hydride superconductors have been the focus of the present field. The superconducting gap characteristics in high-pressure hydride superconductors has been a crucial issue. However, due to the challenges imposed by high-pressure experimental conditions, the conclusive experimental evidence is lacking now. In this work, the authors determine the superconducting gap characteristics of H_3S and D_3S employing high-pressure tunneling spectroscopy, providing insights into the superconducting mechanism of hydride superconductors. However, I think the present data quality is not enough to prove the conclusions, and the present version requires major revisions. Several important questions or aspects regarding the data and viewpoints presented in the manuscript should be confirmed and modified before the final publication:”

We thank the referee for reviewing our work. We appreciate the important questions raised by the referee. Below, we provide detailed responses to the referee's questions and comments.

1. How can the authors ensure sufficient contact between the sample and the electrodes under high pressure to achieve reliable electrical transport and tunneling spectra signals? If the S sample and electrodes are in close contact, it will indicate that there is minimal hydrogen content between them before heating. In this condition, how to ensure that the sample fully reacts to form HxS after heating? Alternatively, if the sulfur sample is fully enveloped by hydrogen, how to achieve high-quality electrical transport and tunneling spectra signal?

We thank the referee for raising this point. In our experiment, we put a small piece of sulfur directly onto the electrical contacts before gas loading. After the gas loading, the main amount of H_2 is above the loaded sample – within the space between sample and the second diamond anvil (the one without electrical leads), since the sample remains laying on the contacts deposited on the first anvil. However, there may be imperfect contact between the sulfur sample and the electrodes before laser heating, because small amount of hydrogen may penetrate between them after the gas loading procedure. However, this small amount of hydrogen is absorbed by the sulfur sample during laser heating, and then sufficient contact is made between the hydride sample and the electrodes, mainly due to substantial expansion of the sample caused by hydrogenation (the molar volume expansion ($S \rightarrow H_3S$) is about 1.65 times). The small amount of hydrogen between electrical leads and sulfur is insufficient for the entire sulfur sample to form pure H_3S , but there is a large amount of hydrogen on the top and side of the sulfur sample that penetrates inside the sample during heating and reacts to form H_3S , as evidenced by the XRD data. The resistivity and tunneling spectra data confirm the formation of continuous high-Tc phase of H_3S which extends to all contacts. Importantly, this technique of the synthesis of metal hydrides and H_3S/D_3S under high pressure

in DACs using H₂/D₂ gas for subsequent electrical resistance measurements has become a routine practice and has been successfully applied for S-H, La-H, Y-H and Ce-H systems.

In the revised version, we have marked this small part of the hydrogen in Fig. 1(a) and addressed this point with the sentence “Note that there may be imperfect contact between the sulfur and the electrodes in samples loaded with H₂ or D₂ before laser heating, because a small amount of H₂ or D₂ may penetrate between them after the gas loading procedure. However, sufficient contact is made between the hydrogenated sample and the electrodes after laser heating due to the expansion after hydrogenation of the sulfur sample.” in the Methods section.

2. In Fig. 2c, the manuscript attributes the asymmetry of the differential conductance curves to an asymmetric metal-oxide tunneling barrier. Does this imply that the Ta₂O₅ film has poor uniformity?

We thank the referee for raising this point. The asymmetry of differential conductance has been explained by Brinkman et. al, considering the tunneling process through asymmetric barrier potential e.g. trapezoidal shape barrier [1]. This asymmetric behavior has been experimentally studied in some metal-oxide barriers such as Ta oxide, Nb oxide and Al oxide [2], and organic impurities in the oxidize layer have been proposed to account for the asymmetry [1-2]. While, the asymmetry of quasiparticles could have different reason which we missed the appropriate reference for this point in the previous version. The asymmetry of quasiparticle peaks has been discussed by Hirsch [3]. He proposed that the energy-dependent transmission of tunneling electrons can lead to the asymmetry of quasiparticle peaks and simulated the influence of this on the tunneling conductance in Sec. III of [3]. Although he claimed that this energy-dependent transmission cannot lead to the opposite sign of the asymmetry in the STM data of the cuprate sample, because the transmission coefficient for an electron does not decrease as its energy increases, it could happen in conventional tunnel junction when the Fermi level is in the band gap of the insulating barrier close to the valence band, as he mentioned. In the revised version, we have included this reference and revised the description of the asymmetry of quasiparticle peaks.

3. They used the Dynes model to fit the tunneling spectrum data in Fig. 3. What advantages does this model offer over the BTK model? I noticed that the barrier parameters are not fitted in this model. Does this make the fitting results less reliable compared to the BTK model?

We thank the referee for raising this point. The Dynes model describes the tunneling density of states through a N/I/S tunnel junction and is widely used in fitting superconducting tunneling spectroscopy, while the BTK model accounts for the transport phenomenon at the N/S interface, incorporating both electron tunneling and Andreev reflections. The main difference between the Dynes model and the BTK model is the barrier strength Z , which characterizes the interface: Z near 0 corresponds to dominant Andreev reflection, $Z > 5$ to a tunneling regime, and intermediate values to a mixture of both. In general, the Dynes model approximates the BTK model in the tunneling limit. In our study, we observed a negligible difference between fitting results with the BTK model in a tunneling limit ($Z=10$) and the Dynes model, as shown below. To ensure straightforward interpretation and minimize the influence of numerical simulations, we prioritized using models with fewer fitting parameters while adequately capturing the underlying physical phenomena. Therefore, we opted for the Dynes model. We also acknowledge that the

temperature effect was not incorporated into the Dynes fitting in the previous version of our manuscript. In the revision, we have addressed this by convolving the superconducting density of states with the energy derivative of the Fermi-Dirac distribution.

Fig. R1-1. Comparison between Dynes (left) and BTK (right) fitting of the tunneling spectrum of $\text{H}_3\text{S-S1}$ at 20 K.

4. Based on the electrical curves in Fig. 2, zero resistance can be maintained up to 190 K. But why does the tunneling spectrum signal exhibit such significant distortion due to thermal effects above 130 K?

We appreciate the referee for raising this point. We did not observe significant distortion in the tunneling spectrum above 130 K; however, our previous description of the loss of energy resolution at higher temperatures may not have been sufficiently clear. Here we provide a more detailed explanation. The energy resolution of tunneling spectroscopy is in the order of a few times the thermal energy [4] and can be estimated as $\Delta E \approx [(3k_B T)^2 + (2.5 \text{ eV}_{\text{mod}})^2]^{1/2}$ [5]. The first term is due to the thermal smearing effect of the Fermi-Dirac function, while the second term results from the modulation delta current that we use to measure the differential conductance. Although the influence of the second term can be minimized by using the small delta current, the thermal effect cannot be avoided. Consequently, the energy uncertainty ΔE becomes significantly larger than the superconducting gap value at temperatures above 130 K. Therefore, we assert that it is not possible to extract reliable gap information above 130 K due to the loss of energy resolution. In the revised version, we have modified the main text and supplemented the tunneling spectrum above 130 K in SI (Fig. S4).

5. The XRD patterns indicate good powder diffraction rings, suggesting that the sample have low crystallinity and thus exhibits isotropic properties. Could this result in the broadening of the differential conductance curve, a U-shaped curve with s-wave gap characteristics?

We thank the referee for raising this point. As shown below, the tunneling channels in a planar tunnel junction are arranged in parallel rather than in series. A series arrangement may cause the bias voltage to spread due to additional resistance in the circuit, but a parallel arrangement should not. The tunneling conductance of the planar junction is a sum of tunnelling conductance from different small poly crystals with different orientations in the junction area. If there is a significant gap anisotropy from different small poly crystals, there will be additional differential conductance inside the large gap region due to the

contribution of crystals with small gap values, which would distort the U-shape structure instead of broadening the overall structure. In our case, the U-shaped curve suggest different small poly crystals with different orientations in the junction area have similar fully-gapped structure and gap value.

FIGURE REDACTED

Fig. R1-2. Schematic illustration of the tunneling current channels in the N/I/S planar tunnel junction.

6. In Fig. 4b, how is the red dashed background line obtained? Do the authors use the differential conductance curve at the normal-state temperature as the background, or fit a parabolic curve as the background?

We thank the referee for raising this point. The red dashed background is obtained by fitting with BDR model. In the revised version, we have revised the description of this background.

7. In Fig. 5a, a V-shaped differential conductance curve near zero bias voltage is observed, while a U-shaped differential conductance curve is also noted earlier in the manuscript. Is it reliable for the authors to determine the superconducting pairing symmetry based on the shape of the differential conductance curve?

We thank the referee for raising this point. We agree with the referee that the pairing symmetry cannot be determined merely by the shape of gap. However, our tunneling spectrum of samples S1, S2 and S3 with the high-purity 1m-3m-H₃S phase can be well fitted by the single s-wave Dynes model, suggesting single s-wave symmetry. The "V-shaped" differential conductance found in the mixed phase sample S4 might be due to the inhomogeneous gap distribution in the junction region, which is caused by the poorly synthesized sample with multiple phases. In the revised version we have made our statement about the gap symmetry more modest.

8. In Fig. 5, the authors mention that the sample is mixed phases. How to rule out the possibility of sample layering, which could lead to the observation of a Josephson signal between the two sample layers?

We thank the referee for raising this point. As the referee mentioned, we could not rule out sample layering in a mixed phase sample. In fact, before developing the tunneling spectroscopy technique, we have observed some Josephson signals from weak links between sample layers in inhomogeneous sample, including multiple Andreev reflections and zero bias peaks, in point contact measurements without insulating barrier. However, the resistance contribution of these weak links is typically on the order of ~Ohm, which is several orders smaller than the tunnel junction resistance, suggesting that it has a

negligible contribution in the circuit. Indeed, we didn't observe any significant Josephson signal contribution (zero bias peaks and multiple Andreev reflections) in the spectrum of sample S4, suggesting that the N/I/S tunneling is dominant in sample S4.

9. The authors should supplement the evolution of the superconducting gap with magnetic field at different temperatures. This will give the information of the superconducting gap dependence on the magnetic fields.

We thank the referee for raising this point. It is difficult to study the superconducting gap evolution with magnetic field in the 9 T facility because the upper critical field of H₃S is quite large (around 90 T near 0 K and more than 60 T below 100 K) [6]. Here we have fitted the spectrum of sample S3 with a modified Dynes model (considering Zeeman splitting), $N(E) = \frac{N_0}{2} \text{Re} \left[\frac{(E+i\Gamma+\mu_B B)}{\sqrt{(E+i\Gamma+\mu_B B)^2 - \Delta^2}} + \frac{(E+i\Gamma-\mu_B B)}{\sqrt{(E+i\Gamma-\mu_B B)^2 - \Delta^2}} \right]$, where Δ and $\mu_B B$ are the gap value and Zeeman energy under magnetic field, respectively, Gamma is the broadening parameter which includes intrinsic quasiparticle recombination and extrinsic inelastic scattering. As shown in the figure below, there is a slight influence of the gap value with 9 T at 2 K. At high temperatures, the magnitude of the loss of energy resolution will be larger than the gap evolution within the available range of magnetic fields up to 9 T (assuming the square root field dependence [7]) with increasing temperature. Therefore, the effective way to study the superconducting gap evolution under magnetic field is to perform tunneling measurements at low temperatures and much higher magnetic fields. These experiments require special facilities and modification of our standard DACs and could be done in the future as a separate project. In the revised version, we have added this figure in SI as Fig. S6 and mentioned in the main text that further studies at high magnetic fields are desired.

Fig. R1-3. Tunneling spectrum of H₃S-S3 at 2 K (blue data points) and simulation with Dynes model (red curve), under external magnetic fields 0 T (a) and 9 T (b). (c) Gap evolution under magnetic fields (red points) and comparison with square root field dependence (black curve). Error bars are from fitting error of Dynes model.

10. The authors claimed they got superconducting gap value of 30 meV for H₃S, but this value is much smaller than the predicted one (PRB 2015, 91, 060511(R)). How to explain this large discrepancy? In general, this method would give the higher superconducting gap due to the broaden of the tunneling caused by the scattering or thermal effects.

We thank the referee for raising this point. As the third referee mentioned, there are two main reasons for the relatively smaller gap value than the theoretical estimation. The first is the trivial extrinsic reason, which is due to the uncertainty from estimating the gap value with Dynes model. The second intrinsic reason is the lack of consideration of some possible non-trivial effects in the theoretical model, because the estimation of gap value is more sensitive to different theoretical approximations (e.g. harmonic/anharmonic; BCS/Eliashberg/SCDFT; isotropic/anisotropic) than the estimation of T_c. For example, the gap value estimated with the anharmonic feature considered [8] is smaller than that without it [9]. Additionally, a possible non-trivial effect that the theoretical approach did not consider in estimating the gap value is, as proposed by the third referee, the feedback effect of superconducting condensation, which could modify the phonon spectrum at low temperature.

References:

- [1] Brinkman W F, Dynes R C, Rowell J M. Tunneling conductance of asymmetrical barriers[J]. Journal of applied physics, 1970, 41(5): 1915-1921.
- [2] Brunner M, Ekrut H, Hahn A. Metal-oxide-metal tunneling junctions on Ta and Nb: Background conductivity resulting from different oxide barriers[J]. Journal of Applied Physics, 1982, 53(3): 1596-1601.
- [3] Hirsch J E. Slope of the superconducting gap function in Bi₂Sr₂CaCu₂O_{8+δ} measured by vacuum tunneling spectroscopy[J]. Physical Review B, 1999, 59(18): 11962.
- [4] Wolf E L. Principles of electron tunneling spectroscopy (Introduction chapter). (Oxford University Press: Oxford, 2012)
- [5] Wittich G. Scanning Tunneling Microscopy and Spectroscopy at Low Temperatures: development of a 1 K-Instrument and Local Characterization of Heterogenous Metal Systems, PhD thesis, 2005.
- [6] Mozaffari S, Sun D, Minkov V S, et al. Superconducting phase diagram of H₃S under high magnetic fields[J]. Nature communications, 2019, 10(1): 2522.
- [7] Brandt E H. Microscopic theory of clean type-II superconductors in the entire field-temperature plane[J]. physica status solidi (b), 1976, 77(1): 105-119.
- [8] Errea I, Calandra M, Pickard C J, et al. High-pressure hydrogen sulfide from first principles: a strongly anharmonic phonon-mediated superconductor[J]. Physical review letters, 2015, 114(15): 157004.
- [9] Bernstein N, Hellberg C S, Johannes M D, et al. What superconducts in sulfur hydrides under pressure and why[J]. Physical Review B, 2015, 91(6): 060511.

Referee #2 (Remarks to the Author)

“Results:

Du et al. present electron tunnelling experiments on the high-temperature superconductors H3S and D3S. The study observes a gap in the electronic states consistent with the expectations for superconductors. The spectra show coherence peaks which are a fingerprint of superconductivity and a U-shaped profile indicating fully gapped superconductivity. The magnitude of the gap is an exact match with the expected value from weak-coupling BCS theory and the difference in gap magnitude for H3S and D3S is consistent with the isotope effect expected for electron-phonon superconductors.

Originality:

This is the first reliable report of electronic spectroscopy in the whole class of hydride high-temperature superconductors. Previous related work was limited to small signatures (~2%) of Andreev reflection on top of a complex background in CeH9 [1]. Other previous work by some of the co-authors on optical reflectivity was obscured by background contributions and is not generally accepted as a reliable [2]. The present work by Du et al. is successfully analysed as 100% quasiparticle tunnelling.

Significance:

The reliable observation of the superconducting gap in this report is highly significant for the field of hydride superconductivity and beyond. Since the discovery of high-temperature superconductivity in H3S, the whole field of hydride superconductivity has been under close scrutiny. In fact, the dialogue has left many observers in doubt whether superconductivity is the correct interpretation. In addition, academic integrity standards of the high-pressure community have been questioned.

Robustness of Conclusions:

The present study provides very solid evidence that superconductivity is the correct interpretation of the signatures observed in H3S and D3S. The observation of a gap in the electronic states combined with the drop in resistance observed here on the same samples rules out virtually all alternative interpretations. In addition, the observed coherence peaks are a fingerprint of superconductivity.

Data and Methodology:

The present manuscript presents highly reproducible data on multiple samples. If this can be complimented by an immediate and full publication of the raw data, it will also rebuild trust in academic integrity. I further

encourage the authors to indicate the total number of pressure cells and samples prepared for tunnelling experiments and what variation of results has been observed in that wider set if applicable.

Given the wide significance and novelty of the results, I recommend publication of the manuscript in Nature once the authors have considered my comments.

I list a few more specific comments:”

We thank the referee for reviewing our paper and recommending it for publication in Nature. We appreciate the referee's accurate and comprehensive overview of our work from various perspectives, as well as his/her constructive suggestions and comments. We respond to the points raised by the referee in detail below.

1. The authors use the Brinkman-Dynes-Rowell model to analyse the background and the Dynes model to analyse the superconducting gap. These models are appropriate.

We thank the referee for raising this point. We agree with the referee on this point.

2. The authors write that they “simulated” the Dynes model. Do the authors mean that they fitted the Dynes model to the data using a least-squares fit or similar? If not, they should explain how the uncertainties on Δ and Γ (Fig. 3b) have been obtained.

We appreciate this point raised by the referee. As the referee mentioned, we fitted the Dynes model to the data using least-square method. In the revised version, we have substituted “simulated” with “fitted”.

3. In the analysis of the Dynes model, the authors include quasiparticle broadening (Γ). The authors should clarify what origin they associate with the broadening as the original work by Dynes linked this to intrinsic quasiparticle recombination whilst in the present study it appears more likely to be linked to extrinsic factors.

We thank the referee for raising this point. According to Dynes' original work, quasiparticle recombination is related to intrinsic electron-phonon coupling. In our study, there is another extrinsic factor, inelastic scattering due to barrier imperfections, that contributes to the quasiparticle broadening. We note that we didn't include the temperature effect on the Dynes fitting in the previous version. In the present version, we have revised it by convolving superconducting density of state with the energy derivative of the Fermi-Dirac distribution.

4. The authors link the U-shape of the spectra to single-gap s-wave superconductivity. Indeed, the U-shape provides evidence for fully gapped superconductivity which, however, can be realised with other order parameters. Thus, the authors should clarify this argument. The authors might want to rephrase

statements like “directly reveal the superconducting gap structure” which suggests k-space resolved spectroscopy.

We thank the referee for raising this point. We agree with the referee that full gap doesn't necessarily lead to single s-wave symmetry. However, our fully gapped tunneling spectrum can be well described by the single s-wave Dynes model, suggesting the single s-wave symmetry. In the revised version, we have made our statement about gap symmetry more modest. We agree with the referee that tunneling spectroscopy is not a k-space resolved spectroscopy like ARPES and cannot "directly reveal" the superconducting gap structure. In the revised version, we have replaced "directly reveal" with "characterize".

5. Fig 2: To provide a reliable reference measurement, the authors must include a curve of the background in the normal state, i.e. well above 200K.

We thank the referee for raising this point. In the revised version, we have added the data measured at 220 K to Fig. 2 and moved the data at 190 K to Fig. S4.

6. Fig2: The authors omit curves between 140K and 180 K – inside the superconducting state. I encourage the authors to include this data in the supplementary information for completeness even if the data cannot be analysed with the Dynes model due to large broadening.

We thank the referee for raising this point. In the revised version, we have added the data measured at temperatures between 140 K and 180 K in Fig. S4.

7. The background shows a large temperature dependence for sample S1 but not S4. The authors should discuss the origin of the temperature dependence.

We thank the referee for raising this point. The temperature dependence of the background comes from the temperature dependence of the tunneling conductance in the normal state, i.e. metal/insulator/metal tunneling. The temperature dependence of normal metal tunneling has been observed experimentally in some normal metal tunnel junctions, such as Al/AlOx/Al [1-2]. Simmons has theoretically studied the thermal influence on electron tunneling by considering the thermal smearing of the Fermi-Dirac function in terms of tunneling current density [3], which basically leads to the increase of tunneling conductance with temperature. Some other theoretical models have been proposed to discuss the thermal influence on the tunneling conductance, such as thermal fluctuation induced tunneling [4]. Additionally, such factors as temperature dependence of the dielectric constant of the barrier, the thermal expansion of the barrier, and the thermal activation across the barrier can also influence the tunneling conductance at high temperatures. In the revised version, we have added the discussion on the temperature dependence of background in the Methods section.

8. The author suggest thermal drift as a source of reduced energy resolution. However, the method indicates use of a PPMS which typically achieves temperature stability of 1mK over the full temperature range of this study. The authors should clarify this.

We thank the referee for raising this point. The term "thermal drift" is not appropriate here. In the revised version, we have replaced it with "thermal smearing". The energy resolution of tunneling spectroscopy is on the order of a few times the thermal energy [5] and can be estimated as $\Delta E \approx [(3K_B T)^2 + (2.5eV_{\text{mod}})^2]^{1/2}$ [6]. The first term is due to the thermal smearing effect of the Fermi-Dirac function, and the second term is due to the modulation delta current that we use to measure the differential conductance. Although the influence of the second term can be minimized by using the smaller delta current, the increase of thermal smearing leads to the loss of energy resolution at high temperatures.

9. The data show a large increase of noise with increasing bias and increasing temperature for sample S1 and S4 but not for samples S2 and S3. The authors should comment on the origin of this noise.

I encourage the authors to include key characteristics of the setup in the methods or supplementary information. This should include the resistance of the tantalum strip and electrodes as this will allow readers to understand the methodology and expected noise level from Johnson noise and shot noise. Can the authors include details on the magnitude of the delta current used?

We thank the referee for raising this point. There are three main sources of noise in the measurement of differential conductance of tunnel junction. One is Johnson-Nyquist noise [7-8], which stems due to thermal fluctuations and proportional to temperature. The other two sources are Flicker noise [9] and shot noise [10-11], which are due to resistance fluctuations that generate a fluctuating voltage in the presence of a constant current, and the discreteness of charge carriers, respectively. Since we use the same modulation delta current over the entire temperature and bias range for the same sample, Flicker noise does not have a significant temperature and bias dependence. However, the Johnson noise increases with increasing temperature, and the shot noise increases with increasing current in the circuit at high bias, which explains the relatively poor signal-to-noise ratio at high temperature and high bias in samples S1 and S4. On the other hand, the junction resistance at normal state of S2 (~200K ohms) and S3 (~250K ohms) is larger than that of S1 (~50K ohms) and S4 (~10K ohms), and there is less tunnel current in the circuit at the same bias voltage in S2 and S3 than in S1 and S4. Therefore, there is less shot noise in S2 and S3 at the same high bias range.

Following the referee's suggestion, we have added the resistance of the tantalum strip and electrodes, the delta current, and the discussion on data noise in the Methods section of the revised version.

10. For sample S1 and S4, the authors observe asymmetry of the quasiparticle peaks whilst this is absent for S2 and S3. The authors suggest this to originate from an asymmetry of the barrier referring to [3,4]. However, I cannot find any model or explanation in these references that is able to make this link. In fact, the asymmetry of the junction is accounted for in the background fit using the BDR model as a horizontal shift of the minimum and appears to be small for S1. Thus, I ask the authors to clarify this argument.

We thank the referee for raising this point. We missed the appropriate reference for this point in the previous version. The asymmetry of quasiparticle peaks has been discussed by Hirsch [12]. He proposed that the energy-dependent transmission of tunneling electrons can lead to the asymmetry of quasiparticle peaks and simulated the influence of this on the tunneling conductance in Sec. III of [12]. Although he claimed that this energy-dependent transmission cannot lead to the opposite sign of the asymmetry in the STM data of the cuprate sample, because the transmission coefficient for an electron does not decrease as its energy increases, it could happen in conventional tunnel junction when the Fermi level is in the band gap of the insulating barrier close to the valence band, as he mentioned. In the revised version, we have included this reference and revised the description of the asymmetry of quasiparticle peaks.

11. The authors also present evidence for a new H-depleted H_xS compound and associate this with the use of ammonia borane as hydrogen donor. The strict association with ammonia borane appears to contrast earlier work of some of the co-authors and others who have reported phase-pure H₃S synthesised from sulphur and ammonia borane [5–7]. The authors may want to comment on this.

We thank the referee for raising this point. The formation of a particular phase depends mainly on two factors: (i) the pressure (P) and temperature (T) conditions of the synthesis and (ii) the presence of the required amount of H₂. P and T are parameters that are relatively easy to control, but the amount of H₂ available for the reaction is difficult to ensure. Therefore, we always try to have an excess of H₂ - in this case, we have enough H₂ to complete the hydrogenation reaction, yielding the product with the highest H content. Having enough or excess H₂ for the hydrogenation reaction is always a challenge and can only be realized by trial and error, therefore not every loading/synthesis is successful. In the case that after the synthesis we can still find the presence of free H₂ (easily detectable by Raman spectroscopy), we have a pure product of the 1m-3m-H₃S phase, which is the highest H-content in the H-S system at about 150 GPa measured so far. Theoretical predictions also do not suggest the higher H content. However, when H₂ is deficient, different products can be formed. The lack of hydrogen can be realized in different scenarios: compression of H₂S gas, laser heating treatment of S+H₂, S+paraffin oil, S+NH₃BH₃ samples at high pressures, etc. Some of such H-depleted phases have been found and described, but systematic structural studies are lacking because it is impossible to precisely control the ratio between S and H₂ in DAC.

Since the measured T_c is significantly lower than about 200 K for the 1m-3m-H₃S phase, we assigned the new phase to the H-depleted phase. We didn't strictly assign the new H-depleted phase to ammonia borane. The pure H₃S phase in reference [13-14] is synthesized with NH₃BH₃/sulfur/NH₃BH₃ sandwich structure, which can provide enough hydrogen source for complete reaction (in some regions over the sample we found H₂ vibration in Raman spectra). The hydrogen sulfide sample synthesized in reference [15] is a mixture of H₃S phase and beta-Po sulfur phase. In our previous experiments, we did not always observe a new H-depleted phase synthesized with ammonia borane, in most cases it is a mixture of 1m-3m phase and sulfur. However, a few new phases have been synthesized with sulfur and paraffin oil or compressed H₂S. The variation of synthesis conditions including pressure and temperature may also play a role in the formation of a particular phase. In the revised version, we have removed the sentence “the sample synthesized using S+NH₃BH₃ precursors....” in the “Superconducting gap of multi-phase hydrogen sulfide” section to avoid the misleading emphasis on ammonia borane.

12. The authors claim that the suppression of the coherence peaks in a magnetic field of 9T confirms the superconducting origin. Whilst there is a transfer of density of states from the coherence peaks into the gap region, it appears that the centre of the coherence peaks shifts to higher energy. Can the authors comment on this aspect? Can the authors extract a magnitude of the gap and compare this with the expected suppression in magnetic field?

We thank the referee for raising this point. There are several possible reasons for the suppression and broadening of coherence peaks and increase of in-gap density of states in an external magnetic field: reduction of quasiparticle lifetime which includes orbital depairing [16-17], spin-orbital scattering [16-17], spin-flip scattering [18] and extrinsic inelastic scattering; Zeeman splitting of the spin states; quasiparticle excitation in the mixed state [19]; and Doppler shift effect induced by the supercurrent outside the vortex cores [20]. Here we use modified Dynes model (considering Zeeman splitting) to fit the 9 T data, $N(E) = \frac{N_0}{2} \text{Re} \left[\frac{(E+i\Gamma+\mu_B B)}{\sqrt{(E+i\Gamma+\mu_B B)^2 - \Delta^2}} + \frac{(E+i\Gamma-\mu_B B)}{\sqrt{(E+i\Gamma-\mu_B B)^2 - \Delta^2}} \right]$, where Δ and $\mu_B B$ are the gap value and Zeeman energy under magnetic field, respectively, Gamma is the broadening parameter which includes intrinsic quasiparticle recombination and extrinsic inelastic scattering. The obtained gap value at 9 T is 30.5 meV, which is slightly smaller than the value measured at 0 T. The slight difference between gap values at 0 T and 9 T follow the square root field dependence [21] (with H_{c2} around 90 T [22]). However, further experiments of tunneling spectroscopy at much higher magnetic fields are desired to study the gap evolution under magnetic fields.

In the revised version, we have added this figure in SI as Fig. S6.

Fig. R2-1. Tunneling spectrum of H₃S-S3 at 2 K (blue data points) and simulation with Dynes model (red curve), under external magnetic fields 0 T (a) and 9 T (b). (c) Gap evolution under magnetic fields (red points) and comparison with square root field dependence (black curve). Error bars are from fitting error of Dynes model.

13. I suggest to specify the direction of magnetic field relative to the planar junction.

We thank the referee for raising this point. The direction of magnetic field is out-of-plane direction. In the revised version, we have provided the direction of magnetic field in the caption of Fig. S6.

14. I recommend the authors to compare their work to that of Cao et al who observed Andreev reflection in CeH9 [1].

We thank the referee for raising this point. In the revised version, we have added the description of Cao et al.'s work and cited the paper.

15. Finally, I wonder whether the authors want to consider a more specific title potentially indicating the experimental nature of the work.

We thank the referee for raising this point. Following the referee's suggestion, we have changed the title to "Superconducting gap of high temperature superconductor H₃S measured by tunneling spectroscopy" in the revised version.

References:

- [1] Das V D, Jagadeesh M S. Tunneling in Al/Al₂O₃/Al MIM structures[J]. *physica status solidi (a)*, 1981, 66(1): 327-333.
- [2] Patiño E J, Kelkar N G. Experimental determination of tunneling characteristics and dwell times from temperature dependence of Al/Al₂O₃/Al junctions[J]. *Applied Physics Letters*, 2015, 107(25).
- [3] Simmons J G. Generalized thermal J-V characteristic for the electric tunnel effect[J]. *Journal of Applied Physics*, 1964, 35(9): 2655-2658.
- [4] Sheng P, Sichel E K, Gittleman J I. Fluctuation-induced tunneling conduction in carbon-polyvinylchloride composites[J]. *Physical Review Letters*, 1978, 40(18): 1197.
- [5] Wolf E L. Principles of electron tunneling spectroscopy (Introduction chapter). (Oxford University Press: Oxford, 2012)
- [6] Wittich G. Scanning Tunneling Microscopy and Spectroscopy at Low Temperatures: development of a 1 K-Instrument and Local Characterization of Heterogenous Metal Systems, PhD thesis, 2005.
- [7] Johnson J B. Thermal agitation of electricity in conductors[J]. *Physical review*, 1928, 32(1): 97.
- [8] Nyquist H. Thermal agitation of electric charge in conductors[J]. *Physical review*, 1928, 32(1): 110.
- [9] Voss R F, Clarke J. Flicker (1/f) noise: Equilibrium temperature and resistance fluctuations[J]. *Physical Review B*, 1976, 13(2): 556.
- [10] Birk H, De Jong M J M, Schönenberger C. Shot-noise suppression in the single-electron tunneling regime[J]. *Physical review letters*, 1995, 75(8): 1610.

- [11] Blanter Y M, Büttiker M. Shot noise in mesoscopic conductors[J]. Physics reports, 2000, 336(1-2): 1-166.
- [12] Hirsch J E. Slope of the superconducting gap function in $\text{Bi}_2\text{Sr}_2\text{CaCu}_2\text{O}_{8+\delta}$ measured by vacuum tunneling spectroscopy[J]. Physical Review B, 1999, 59(18): 11962.
- [13] Minkov V S, Ksenofontov V, Bud'ko S L, et al. Magnetic flux trapping in hydrogen-rich high-temperature superconductors[J]. Nature Physics, 2023, 19(9): 1293-1300.
- [14] Minkov V S, Bud'ko S L, Balakirev F F, et al. Magnetic field screening in hydrogen-rich high-temperature superconductors[J]. Nature Communications, 2022, 13(1): 3194.
- [15] Osmond I, Moulding O, Cross S, et al. Clean-limit superconductivity in Im_3m H3S synthesized from sulfur and hydrogen donor ammonia borane[J]. Physical Review B, 2022, 105(22): L220502.
- [16] Meservey R, Tedrow P M, Bruno R C. Tunneling measurements on spin-paired superconductors with spin-orbit scattering[J]. Physical Review B, 1975, 11(11): 4224.
- [17] Worledge D C, Geballe T H. Maki analysis of spin-polarized tunneling in an oxide ferromagnet[J]. Physical Review B, 2000, 62(1): 447.
- [18] Grimaldi C, Fulde P. Nonequilibrium superconductivity in spin-polarized superconducting tunneling junctions[J]. Physical Review B, 1997, 56(5): 2751.
- [19] Hess H F, Robinson R B, Dynes R C, et al. Scanning-tunneling-microscope observation of the Abrikosov flux lattice and the density of states near and inside a fluxoid[J]. Physical review letters, 1989, 62(2): 214.
- [20] Kohen A, Proslie T, Cren T, et al. Probing the superfluid velocity with a superconducting tip: the doppler shift effect[J]. Physical review letters, 2006, 97(2): 027001.
- [21] Brandt E H. Microscopic theory of clean type-II superconductors in the entire field-temperature plane[J]. physica status solidi (b), 1976, 77(1): 105-119.
- [22] Mozaffari S, Sun D, Minkov V S, et al. Superconducting phase diagram of H3S under high magnetic fields[J]. Nature communications, 2019, 10(1): 2522.

Referee #3 (Remarks to the Author)

"Results:

I have reviewed the paper "Superconducting gap of high temperature superconductor H3S" by F. Du and coworkers.

In this paper is reported new data on the synthesis and characterization of sulphur hydride at extreme pressure with a particular focus on the

measurement of the superconducting gap. This work confirms the observation of high T_c superconductivity in the 1m-3m structural phase of H3S

and provides and measures a superconducting gap of about 30meV (reduced to 22meV with the isotopic substitution H to D).

The impression is that this work presents the most precise and complete characterization of H3S which I have seen so far.

I think that this is relevant because there have been controversial claims on other high pressure hydrides and the reproducibility (/improvement) of earlier measurements in H3S is a significant fact. The paper is clear and the methodology appears to be cutting-edge. However, as I have a theoretical background, I will not comment on the experimental aspects of this work.

From a theoretical point of view there is no (reasonable) doubt about the phononic nature of superconductivity in H3S. In fact superconductivity was predicted before the experiments. The agreement between theoretical and experimental T_c is also quite good in all the many published works (I should add: within the typical theoretical/experimental errorbar).

In fact, the agreement is almost too good, because different methods and approximations all lead to very similar values of T_c . For example it is generally accepted that anharmonic effects are important in this class of materials, as these have a major effects on the lattice dynamics. However even in the harmonic approximation T_c estimations are reasonable.

It nails down to the fact that T_c is not the most sensitive property of phononic superconductors.

The superconducting gap is different. Its estimation can be affected by the use of different theoretical approximation (e.g. harmonic/anharmonic; BCS/Eliashberg/SCDFT ; isotropic/anisotropic). An accurate

estimation of the superconducting gap is, for the theory community, a very valuable element to test the methods and possibly to improve them.

Clearly there are thousand of superconducting systems where these theories can be tested, but none of them has the properties of H3S. H3S is the most extreme available data point in terms of T_c and characteristic phonon frequency.

In this work it is reported that the experimental values of the gap are quite lower than in the theoretical simulations. I would place the most precise theoretical value for H3S to about 36K using Eliashberg theory and anharmonic phonons. The disagreement with the experimental estimation of 30meV is certainly not large enough to doubt of the phononic mechanism, but it indicates a problem that, as the authors mention in the conclusion, should be addressed.

The explanation could be as trivial as a failure of the Dyson model used for the fitting and the comparison with an isotropic (clean limit) estimation of the gap. But this data could also be pointing to the necessity of including non trivial effects to our methods.

One thing that comes to my mind is the feedback effect of the superconducting condensation. In a system with an extremely high gap, its presence could modify the phonon spectrum at low temperature.

This mechanism would affect the theoretical estimation of the gap, without affecting the estimation of T_c (being the transition of the II order the gap is zero at T_c).

In conclusion I have a good opinion of this work. An accurate confirmation of the SH3 measurements is certainly welcome. Most importantly I think that this work provides extremely valuable data which could be quite useful for the improvement of superconductivity theory.

We thank the referee for reviewing and recommending our work. We also appreciate the insightful discussion on the difference in gap value between our experiment and theoretical approaches, from which we learned a lot.

List of changes:

1. The title has been changed to “Superconducting gap of high temperature superconductor H₃S measured by tunneling spectroscopy”.
2. There was a typo in the author's name, it has been corrected. An affiliation of the author has been added.
3. In the abstract, the phrase “directly reveal” has been replaced by “characterize”. The sentence about the gap symmetry has been revised.
4. A description of Andreev reflections work has been included in the Introduction, and the related reference has been cited.
5. In para. 2, page 4, the sentence about asymmetry of quasiparticle peak has been revised and related reference has been cited.
6. In para. 3, page 4, the description of normalized tunneling spectrum has been revised and the phrase “thermal drift” has been revised to “thermal smearing”.
7. Descriptions about Dynes model have been moved to the Methods section.
8. In para. 1, page 5, a sentence about gap evolution under magnetic fields has been added.
9. In para. 2, page 5, the description of normalized tunneling spectrum has been revised.
10. In the last paragraph of page 5, the sentence “the sample synthesized using S+NH₃BH₃ precursors....” has been removed.
11. Phrases about gap symmetry in the text have been revised or removed. Phrase “simulated” has been revised to “fitted” in the text.
12. Discussion on contacts between sample and electrodes has been appended in the Methods section.
13. The resistance of the tantalum strip and electrodes and the magnitude of the delta current used have been appended in the Methods section.
14. Discussion on data noise and temperature dependence of background in tunneling spectrum has been appended in the Methods section.
15. In Figure 1 (a), a small amount of hydrogen between sulfur and electrodes have been marked with pink color.
16. In Figure 2(d), data measured at 220 K has been added and data measured at 190 K has been moved to Figure S4.
17. In Figure 3, normalized conductance at 20 K to 130 K have been re-fitted with the Dynes model after considering the temperature effect.

18. Tunneling spectra measured between 140 K to 190 K have been appended to Figure S4.
19. The Dynes fit of the 9 T data from H₃S-S3 and the gap evolution under magnetic fields have been appended to Figure S6.
20. Other minor changes are highlighted in red.

Response to the referees

We sincerely thank all the referees for reviewing our manuscript and for their valuable comments and insightful questions. Below, we provide detailed responses to each of the referees' comments. A list of the corresponding changes made in the revised manuscript follows.

Referee #1 (Remarks to the Author)

"The authors have made some improvements to the paper. However, they have not really answered my queries. I still have several issues regarding this work as follows:"

We sincerely thank the referee for their continued review of our manuscript and for acknowledging the improvements made in the previous version. We regret if any of the queries were insufficiently addressed and appreciate the opportunity to clarify further. Below, we provide detailed responses to the referee's comments and questions.

1. In Supplementary Material Fig. S1, they provided an image of the diamond surface used for high-pressure tunneling spectroscopy measurements. I estimated the width of the Ta line on the diamond anvil surface to be approximately 2–3 microns. For tunneling spectroscopy measurements under ambient pressure, the tip is required to be at the nanometer scale. Do you think a micron-scale contact area can meet the experimental requirements? Will the contact area have a significant impact on the experimental results?

We thank the referee for this thoughtful question. There are two primary approaches to conducting tunneling spectroscopy measurements under atmospheric pressure:

1. **Planar tunnel junctions**, as utilized in this work and described in the manuscript.
2. **Point contact technique**, where the contacts are smaller than the mean-free path of electrons, such as those employed in scanning tunneling microscopy/spectroscopy (STM/STS), where the needle tip and vacuum serve as the metal-insulator part of the tunnel junction.

As the referee noted, STM/STS technique rely on nanometer-scale tips to achieve high spatial resolution. In contrast, planar tunnel junctions typically involve a larger contact area, sacrificing spatial resolution. However, the size of the contact area in planar tunnel junctions does not significantly affect the tunneling process itself.

To support this, we cite several studies that successfully used planar tunnel junctions with contact areas ranging from several square microns to several hundred square microns:

- **Quantum Front. 2, 5 (2023):** <https://doi.org/10.1007/s44214-023-00031-3>
- **Nat Commun 9, 598 (2018):** <https://doi.org/10.1038/s41467-018-03000-w>
- **Phys. Rev. B 95, 195129 (2017):** <https://doi.org/10.1103/PhysRevB.95.195129>
- **Appl. Phys. Lett. 107, 253502 (2015):** <https://doi.org/10.1063/1.4938209>

Notably, the pioneering work by Giaever on tunneling spectroscopy experiment to measure superconducting gap of Pb utilized planar tunnel junctions with a contact width of 1 mm (Al/Al₂O₃/Pb junctions) [Phys. Rev. **122**, 1101 (1961): <https://doi.org/10.1103/PhysRev.122.1101>].

In summary, although the contact area in our experiments is larger than the nanometer scale used in STM/STS techniques, this does not impact the validity of the tunneling spectroscopy measurements performed using the planar tunnel junction approach.

2. I'm still puzzled as to why the tunneling spectroscopy signal decays so rapidly in the temperature range between 130 K and 190 K, even under conditions of zero resistance. In the high-pressure tunneling spectroscopy experiments of elemental sulfur (PRL 133, 036002 (2024)), I didn't observe the similar phenomena. Although the authors mentioned that the signal quality significantly deteriorates above 130 K, according to the curve in Figure 3a, the tunneling signal is already no longer obvious above 110 K. Based on the article in Nature, 525, 73–76 (2015), H₂S exhibits a superconducting transition temperature near 100 K under similar pressure ranges. Therefore, is it possible that the Ta-Ta₂O₅ tunneling junction, spanning the entire sample, is detecting a mixture of signals from H₃S and H₂S? The zero-resistance observed in electrical transport measurements only proves the formation of a superconducting path of H₃S between the two gold electrodes. The powder XRD diffraction pattern provided by the authors only confirms the presence of pure-phase H₃S in one specific region, but it does not cover all the samples within the sample chamber. Thus, the accuracy of the bandgap information detected using this method remains open to discussion.

We thank the referee for their detailed analysis and for raising this important question. We address the referee's concerns in two parts, as outlined below.

1. Purity of the Im-3m-H₃S phase in the sample.

We confirmed the purity of in the Im-3m-H₃S phase in the tunneling junction region through detailed X-ray diffraction (XRD) mapping. To provide stronger evidence, we have now integrated the XRD patterns specifically in the junction region, as shown in the revised version of Supplementary Figure S2.

The data reveal no detectable secondary phases, including H₂S, within this region. This confirms that the detected tunneling signal originates solely from pure Im-3m-H₃S. Furthermore, the excess H₂ gas during the synthesis ensures the complete hydrogenation reaction to form the highest-hydrogen-content product.

To illustrate this, we have provided the following Figure R1-1:

Fig. R1-1. (a) X-ray diffraction map of H₃S-S1. The color brightness varies with the intensity at the characterized diffraction angle around 9.9 degree at different data points. The tunnel junction area is marked with red dashed rectangle. (b) Integrated X-ray powder diffraction pattern of the junction region of H₃S-S1.

These results strongly support the conclusion that the tunneling signal is not influenced by a mixture of H₃S and H₂S phases, as suggested by the referee.

2. Decay of tunneling spectroscopy signal at high temperatures.

The rapid decay of the tunneling spectroscopy signal at higher temperatures can be attributed to several factors intrinsic to tunneling spectroscopy rather than a mixture of H₃S and H₂S.

(a) Thermal smearing effect and loss of energy resolution:

The single-electron tunneling technique works best at low temperatures because tunneling electrons obey the Fermi-Dirac distribution. As temperature increases, the Fermi-Dirac distribution broadens, reducing the energy resolution. Importantly, this smearing increases with absolute temperature rather than relative temperature (e.g., relative to T_c). Thus, at temperatures above ~ 110 K, the gap feature becomes less distinguishable.

(b) Quasiparticle broadening effects:

Both intrinsic (e.g., electron-phonon coupling) and extrinsic (e.g., inelastic scattering due to imperfections in the barrier) broadening effects grow with temperature. These effects exacerbate the smearing of the

superconducting gap feature. Above 110 K, the gap value decreases significantly, as predicted by BCS theory, while broadening effects and energy resolution loss dominate, effectively obscuring the quasiparticle peaks and weakening the signal within the gap region.

(c) Influence of the parabolic background from Ta₂O₅-based tunnel junctions:

In our experiments, the tunneling spectrum exhibits a parabolic background due to normal metal tunneling. At low bias voltage (< 10 mV), this background is nearly constant and does not significantly affect the gap feature: the smeared tunneling spectrum with a small gap value and a weak influence of quasiparticle broadening effects and energy resolution loss (at low absolute temperature) can be discerned, as in the case of [PRL 133, 036002 (2024)]. However, at higher bias voltages (as in our case), the slope of the parabolic background becomes more pronounced. The drop of the smeared tunneling spectrum is even much weaker than the change of the background from high bias to zero bias, rendering the gap feature indistinguishable.

For a more specific explanation of the thermal smearing effect, we simulated the measured tunneling spectroscopy data with a largely smeared gap feature at 150 K. The gap value and broadening parameter (Γ) were set to 20 meV and 30 meV, respectively. The simulated measured data, shown in Figure R1-2 (c) were obtained by multiplying the simulated normalized tunneling data (Figure R1-2 (a)) with the simulated parabolic background (Figure R1-2 (b)).

Fig. R1-2. (a) Simulated normalized tunneling spectrum at 150 K. (b) Simulated parabolic background at 150 K. (c) Simulated measured data at 150 K obtained by multiplying the simulated normalized tunneling data (a) with the simulated parabolic background data (b).

As shown in Figure R1-2(c), the simulated measured data also exhibit a parabolic behavior and can be well-fitted with a parabolic function. However, the fitting parameters differ from those of the background in Figure.R1-2 (b).

Addressing technical limitations.

While our current Ta₂O₅-based tunneling junctions face challenges at high temperatures, future improvements could involve developing barrier materials with reduced background effects or employing high-temperature superconducting probes. However, maintaining performance at megabar pressure conditions remains a significant challenge.

In conclusion, based on experimental data, simulations, and analysis, the rapid decay of the tunneling spectroscopy signal at higher temperatures is attributed to thermal effects and the parabolic background, not to signal mixing from H₃S and H₂S phases, as suggested by the referee.

3. The resistance of an insulator changes exponentially or linearly with temperature, which should result in a faster rate of change at low temperatures. However, why does the contact resistance in the variable-temperature tunneling spectroscopy data presented in the paper show a slower rate of change at low temperatures and a faster rate of change at high temperature range?

We thank the referee for raising this point. Below, we provide an explanation based on both theoretical and experimental findings.

The tunnel conductance/resistance of a tunnel junction is not solely determined by the conductance/resistance of the insulating barrier. The temperature dependence of tunneling conductance has been theoretically studied by Simmons in the context of normal metal junctions [**Journal of Applied Physics**, 1964, 35(9), 2655-2658: <https://doi.org/10.1063/1.1713820>]. According to Simmons' model, the temperature dependence follows a T-square behavior, which agrees with our observations in this work. This behavior explains the referee's observation of a "slower rate of change at low temperature and a faster rate of change at high temperature range".

Experimentally, similar temperature-dependent behavior of tunnel junction resistance and current-voltage (I-V) characteristics has been observed in normal metal-insulator-normal metal tunnel junctions, such as Al/Al₂O₃/Al. For instance, Figure 6.3 in the Master's Thesis "Bias and temperature dependence analysis of the tunneling current of normal metal-insulator-normal metal tunnel junctions" (<https://jyx.jyu.fi/bitstream/handle/123456789/8190/G0000137.pdf>) and studies in *Applied Physics Letters* [**Appl. Phys. Lett.** 107, 253502 (2015): <https://doi.org/10.1063/1.4938209>] support the observed trend of slower resistance changes at lower temperatures and faster changes at higher temperatures.

It should also be noted that there is no universal temperature dependence for the tunnel conductance in a junction, as multiple factors can influence it. These factors include: Temperature dependence of the dielectric constant of the barrier material: Variations in the dielectric properties of the barrier with temperature can modify the tunneling process; Thermal expansion of the barrier: Changes in barrier thickness or structure due to thermal expansion can affect tunneling conductance; Thermal activation across the barrier: At higher temperatures, thermally activated processes can contribute to the conductance.

Based on both theoretical and experimental evidence, the observed behavior in our work is consistent with expectations for tunnel junctions and reflects a combination of these effects.

4. Why do the tunneling spectroscopy curves in Figure 5 exhibit a V-shaped characteristic different from other DACs?

We thank the referee for raising this point. Below, we address the concern systematically:

1. Clarification on pairing symmetry and tunneling spectrum shape.

The pairing symmetry of superconductors cannot be determined directly from the shape of the tunneling spectrum. Such an analysis requires extremely high-purity samples, precise identification of crystal orientation, and measurements at very low temperature (close to zero) with negligible quasiparticle broadening effects. This is why we rephrased the symmetry statement in the previous version of the manuscript.

2. Impact of quasiparticle broadening and thermal smearing effects.

For single s-wave superconductors, large quasiparticle broadening effects – both intrinsic and extrinsic – combined with thermal smearing effects at finite temperatures can distort the U-shaped tunneling spectrum into a V-shaped characteristic. As demonstrated in Fig.R1-3 below, a small gamma value results in a U-shaped structure (Fig. R1-3a), whereas a larger gamma value with the same gap produces a V-shaped characteristic (Fig. R1-3b). Additionally, inhomogeneous gap distributions in poorly synthesized samples could also distort the tunneling spectrum, as we discussed in the previous response.

Fig. R1-3. (a) Simulated tunneling spectrum of single s wave superconductor with moderate Gamma value. (b) Simulated tunneling spectrum of single s wave superconductor with large Gamma value.

3. Exclusion of d-wave pairing symmetry.

The shape of the tunneling spectrum for d-wave superconductors depends on crystal orientation. As shown in Fig. R1-4 (see also Supplementary Information in PRL 133, 036002 (2024)), tunneling spectra exhibit a V-shaped feature only when measured along the anti-nodal direction, whereas a significant zero-bias peak immerses along the nodal direction. For polycrystalline samples, such as those synthesized in our experiments, the spectrum should include contributions from different crystal orientations. Simulation in Fig. R1-4 indicate that a spectrum from d-wave superconductors with averaged contributions from all crystal orientations would exhibit clear zero-bias peak, which is inconsistent with the data of sample 4. Therefore, the tunneling spectrum of sample 4 does not support d-wave pairing.

Fig. R1-4. Simulated tunneling spectrum of d wave superconductor when the vertical interface orientation of the junction is along the nodal direction (red curve), anti-nodal (black curve) and averaged orientations (cyan curve) respectively.

4. Analysis of sample 4's tunneling spectrum with the s-wave Dynes model.

We fit the tunneling spectrum data of sample S4 at 40 K using a two-component single s-wave Dynes model, where $N(E) = \omega \cdot N_s(E) + (1 - \omega) \cdot N_o(E)$. Here, superconducting and non-superconducting components are weighted by ω and $1 - \omega$, respectively. As shown in Fig. R1-5, the tunneling spectrum of sample 4 is well described by the s-wave Dynes model, consistent with samples 1-3. The large gamma value indicates relatively poor barrier quality, while the superconducting fraction $w = 81.9\%$ reflects phase inhomogeneity in sample 4. The slightly smaller gap value aligns with the lower T_c of sample S4 ($T_c^{\text{offset}}(\text{H}_3\text{S}) \approx 175 \text{ K}$).

Fig. R1-5. (a) Tunneling spectrum of sample S4 at 40 K and fit with the BDR model. (b) Normalized tunneling spectrum of sample S4 at 40 K and fit with the two-component Dynes model.

Thus, the observed V-shaped distortion in the tunneling spectrum of sample 4 is due to the combined effects of poor barrier quality and phase inhomogeneity, rather than a difference in pairing symmetry or a deviation from s-wave superconductivity. This is consistent with our findings across multiple samples.

In the revised version, we have:

- Added the sentence “The distortion of the fully gapped structure in the tunneling spectrum of sample 4 is due to the relatively poor barrier quality and phase inhomogeneity, as analyzed in Fig. S7.” to the caption of Figure 5(b).
- Included the analysis of the data at 40 K in sample 4 in Fig. S7.

5. The tunneling spectroscopy curves presented by the authors in the high-pressure region generally have a poor signal-to-noise ratio, while the signal-to-noise ratio is better in the voltage range within the superconducting gap. Did the authors perform any fitting or signal processing on the curves to improve their quality?

We thank the referee for this insightful question. We would like to clarify that, except for the normalized data, all tunneling spectroscopy curves presented in this work are raw data obtained directly from measurements, without any fitting or signal processing applied to improve their quality.

The better signal-to-noise ratio observed within the superconducting gap region can be attributed to the following factors:

1. Reduced shot noise in the low bias region. At lower bias voltages, the level of shot noise is inherently lower, which improves the signal-to-noise ratio in this region.

2. Larger differential voltage signal within the superconducting gap. In our differential conductance measurements, a constant delta current (δI) is applied using a Keithley current source, and the corresponding differential voltage (δV) is measured by a Keithley nanovoltmeter. The differential conductance is then calculated by the instrument as $\delta I / \delta V$. Within the superconducting gap, the differential voltage signal is larger, and noise levels are relatively low, leading to an enhanced signal-to-noise ratio.

Referee #2 (Remarks to the Author)

I recommend publication of the manuscript. The authors have addressed all my queries satisfactorily and I only have a single remaining comment (see below).

I suggest the authors to describe/check carefully, their data fitting routines. The fits shown in Fig. 3a seem much closer to the data for negative bias than for positive bias. I wonder whether only the data at negative bias has been fitted. If so, this should be clearly described and discussed. Whilst I don't expect that this has significant relevance for the main conclusions of the manuscript, the authors may want to be cautious to avoid any ambiguity.

We thank the referee for their recommendation of our work and for this important observation regarding the data fitting routines in Figure 3a. To clarify, the normalized tunneling spectra in Figure 3a were indeed fitted using data at negative bias. This approach was adopted to minimize the influence of energy-dependent transmission in determining the gap value.

In the revised version of the manuscript, we have added a clear description of this fitting routine in the caption of Figure 3a to avoid any ambiguity. We agree with the referee that this does not affect the main conclusions of the manuscript but appreciate the opportunity to ensure clarity in the presentation of our data.

Referee #3 (Remarks to the Author)

I believe the Authors have replied to all the points raised by the referees. I maintain my opinion that this paper is suitable for publication in Nature.

We sincerely thank the referee for reviewing our manuscript and for their recommendation for publication. We greatly appreciate their support and positive evaluation of our work.

List of changes:

1. The integration of the X-ray pattern of the junction area has been added to Figure S2.
2. In para. 3, page 4, the sentence about the tunneling spectra above 130 K has been revised.
3. A description of the distortion of the fully gapped structure in the tunneling spectrum of sample 4 has been added to the caption of Figure 5(b). Additionally, the analysis of the data at 40 K for sample 4 has been incorporated into Figure S7.
4. A description of the data fitting routines has been added to the caption of Figure 3 (a).
5. Other minor changes throughout the manuscript have been highlighted in red.

Third response to the reviewers:

Referee #1 (Remarks to the Author)

“In this revised version, I think the authors have addressed most of my queries. I recommend publication of the manuscript.”

We sincerely thank the referee for reviewing our manuscript and for their recommendation for publication.